# ABHD5 inhibits YAP-induced c-Met overexpression and colon cancer cell stemness via suppressing YAP methylation

Yan Gu[1,4], Yanrong Chen[1,4], Lai Wei[1,4], Shuang Wu[1,4], Kaicheng Shen [1], Chengxiang Liu[1], Yan Dong[1], Yang Zhao[1], Yue Zhang[1], Chi Zhang[1], Wenling Zheng[1], Jiangyi He[1], Yunlong Wang[1], Yifei Li[1], Xiaoxin Zhao[1], Hongwei Wang[1], Jun Tan[1], Liting Wang[2], Qi Zhou[3✉], Ganfeng Xie[1✉], Houjie Liang[1✉] & Juanjuan Ou [1✉]

Cancer stemness represents a major source of development and progression of colorectal cancer (CRC). c-Met critically contributes to CRC stemness, but how c-Met is activated in CRC remains elusive. We previously identified the lipolytic factor *ABHD5* as an important tumour suppressor gene in CRC. Here, we show that loss of *ABHD5* promotes c-Met activation to sustain CRC stemness in a non-canonical manner. Mechanistically, we demonstrate that ABHD5 interacts in the cytoplasm with the core subunit of the SET1A methyltransferase complex, DPY30, thereby inhibiting the nuclear translocation of DPY30 and activity of SET1A. In the absence of ABHD5, DPY30 translocates to the nucleus and supports SET1A-mediated methylation of YAP and histone H3, which sequesters YAP in the nucleus and increases chromatin accessibility to synergistically promote YAP-induced transcription of c-Met, thus promoting the stemness of CRC cells. This study reveals a novel role of ABHD5 in regulating histone/non-histone methylation and CRC stemness.

[1] Department of Oncology and Southwest Cancer Center, Southwest Hospital, Third Military Medical University (Army Medical University), 400038 Chongqing, China. [2] Biomedical Analysis Center, Third Military Medical University (Army Medical University), 400038 Chongqing, China. [3] Department of Oncology, Fuling Central Hospital of Chongqing City, 408000 Chongqing, China. [4] These authors contributed equally: Yan Gu, Yanrong Chen, Lai Wei, Shuang Wu. ✉email: qizhou112@163.com; xiegf@aliyun.com; lianghoujie@sina.com; 13957751@qq.com

Cancers represent a collection of highly heterogeneous malignant diseases that comprise cellular hierarchies defined as sub-populations of cancer stem cell (CSCs) within a majority of more differentiated cancer cells, possessing transcriptional and epigenetic programmes endowing them with stemness properties, such as self-renewing, responsible for cancer pathogenesis[1–3]. Colorectal carcinoma (CRC) usually follows the CSC model, and it has been demonstrated that the stemness of CRC cells critically contributes to tumour initiation, progression and metastasis[4,5]. Despite the central role of CSCs in CRC malignancy, the regulation of CRC stemness remains elusive. Therefore, unlocking the molecular mechanism responsible for CRC stemness is key in the development of novel therapeutics for the total elimination of CSCs thus improving the treatment modalities.

c-Met, the protein product of *MET* proto-oncogene, is a receptor-type tyrosine kinase (RTK) playing an important role both in physiological and pathological processes. In cancer, c-Met promotes tumour angiogenesis, growth and metastasis and has been demonstrated to be an exciting novel drug target[6–9]. c-Met activation in cancer can occur by gene mutation, amplification and overexpression independent of binding to its ligand hepatocyte growth factor (HGF). Recently, c-Met gene amplification is receiving increased attention because c-Met overexpression at mRNA and protein levels have been reported in many varieties of human cancers, including CRC, which has been demonstrated to be critically attributable to CRC stemness and poor prognosis[10,11]. Unmasking the mechanism by which c-Met overexpression is induced in CRC therefore is of great significance for finding out the effective approach for CRC therapy.

Aberrant activation/inactivation of metabolic genes is a hallmark of cancer. Metabolic genes are reported to critically contribute to cancer stemness via metabolic or non-metabolic mechanisms[12–14]. However, the underlying mechanism remains to be further elucidated. We previously identified a/b-hydrolase domain-containing 5 (ABHD5), an intracellular lipolytic activator that is also known as comparative gene identification 58 (CGI-58), as a novel tumour suppressor in CRC[15]. ABHD5 is a well-known cofactor of patatin-like phospholipase domain-containing 2 (PNPLA2, also known as adipose triglyceride lipase, ATGL)[16] that catalysers the first step of lipolysis, converting triglycerides (TGs) to diacylglycerols (DAGs)[17]. ATGL requires ABHD5 to achieve full TG hydrolase activity. Mutations in human *ABHD5* cause Chanarin−Dorfman syndrome[18,19], a rare autosomal-recessive genetic disease characterized by TG-rich LD accumulation in almost all tissues. Mutations in human ATGL also cause a neutral lipid storage disease[20–22]. Despite this similarity, obvious phenotypic differences exist between *ATGL* and *ABHD5* mutations. For example, mice lacking *Abhd5* die neonatally, but mice lacking *Atgl* are viable[21,22], indicating a distinct and ATGL independent role of ABHD5 in embryonic development and differentiation. Our previous studies have revealed a critical role of ABHD5 in suppressing CRC tumourigenesis and metastasis, whether ABHD5 is attributable to CRC stemness remains to be elucidated.

In this study, we further explored the molecular mechanism responsible for the tumour suppressor function of ABHD5 in CRCs and demonstrated that ABHD5 inhibits YAP-induced c-Met overexpression and CRC stemness through suppressing SET1A-induced YAP and histone methylation. This study for the first time reveals an unrecognized role of ABHD5 in controlling the methylation of histone/non-histone proteins, and the subsequent effect on c-Met activation and CRC stemness.

## Results

**Loss of ABHD5 promotes the stemness of CRC cells.** As the loss of ABHD5 promotes tumourigenesis and malignant behaviours of intestinal tumours and CRC cells[15], we speculated that ABHD5 plays a role in regulating the stemness of CRC cells. To investigate this potential role of ABHD5, we used CRC cell lines, which include a small population of cells that molecularly and functionally behave as CSCs. These CRC stem cells were isolated and amplified from human colon cancer cell line HCT116 cells by culturing cloned spheres in serum-free medium supplemented with bFGF and EGF, and analysed for the presence of stem cell markers and tumour-initiating capacity. We analysed how with shRNA-mediated *ABHD5* knockdown affected the ability of HCT116 cells to form spheres in serum-free medium, an indication of stem cell-like behaviour. By using two lentiviral vector-based shRNA to silence the expression and function of ABHD5 in HCT116 cells, we found that both the primary and secondary sphere formation capacity was significantly higher in the *ABHD5*-knockdown group (Fig. 1a, b and Supplementary Fig. 1a–c). Since we have screened the expression levels of ABHD5 in CRC cell lines and found that HCT116 and SW620 showed the relatively highest expression levels of ABHD5[15]. We further silenced *ABHD5* in SW620, and found that silencing *ABHD5* in SW620 also significantly increased their sphere formation capacity as that in HCT116 cells (Supplementary Fig. 1a, b, d, e). Conversely, the overexpression of *ABHD5* significantly decreased the sphere-forming capacity of HCT116 cells (Supplementary Fig. 1f). Anchorage-independent (AI) growth is also an important assay evaluating the stemness of cancer cells. Expectedly, *ABHD5*-knockdown HCT116 cells exhibited more pronounced colony formation in soft agar than control cells (Fig. 1c and Supplementary Fig. 1g, h). Moreover, analysis of stem cell numbers by staining for ALDH, CD133 and CD44, reported markers of CRC stem cells, showed that *ABHD5* knockdown significantly increased the proportions of ALDH⁻, CD133⁻ and CD44⁻ positive cells compared to those in control HCT116 cells (Fig. 1d, e).

Given the in vitro findings of increased CSC number and stemness behaviour, we aimed to validate these findings in vivo in our previously generated vil-Cre-pc$^{fl/fl}$ Abhd5$^{fl/fl}$ mice (Apc$^{Min/+}$/Abhd5$^{f/f/Cre+}$)[15]. After *Abhd5* knockout, intestinal tumours showed a significant increase in the number of Lgr5-GFP-positive cells, characterized as CSCs (Fig. 1f). To further evaluate the effects of *ABHD5* knockdown on CRC stemness, we performed limiting dilution analysis, the gold standard in vivo assay for determining the stemness capacity of cancer cells. NOD/SCID mice were subcutaneously injected with five different dilutions of monolayer cultured HCT116 cells ($1 \times 10^6$ to $1 \times 10^4$) and monitored for tumour growth (Fig. 1g). After 20 days, the xenograft tumours in the group injected with $1 \times 10^6$ cells reached 800 mm³, and all mice were scored for the presence of tumours (volume > 0.1 cm³) as evidence of tumour initiation. At all dilutions, the number of mice that developed tumours was higher in mice injected with *ABHD5*-knockdown HCT116 cells than in mice injected with control cells (Fig. 1h). Calculation of the stem cell frequency in mouse tumours indicated that *ABHD5* knockdown resulted in a 6.8-fold increase in the number of CSCs per tumour compared to that in tumours derived from control cells (Fig. 1h). Additionally, as shown in Fig. 1i, the incidence of tumour formation and the tumour volume was significantly increased in the *ABHD5*-knockdown group relative to the control group at different time points. Consistent with the increased number of stem cells, *ABHD5*-knockdown tumours exhibited more robust growth than control tumours, as indicated by the average final tumour weight (Fig. 1j). Next, the stem cell function of excised and dissociated tumours was analysed by sphere formation assay. Importantly, the increased stem cell frequency was also reflected ex vivo, as *ABHD5*-knockdown cells from the excised tumours displayed increased proliferation and self-renewal (Fig. 1k). Conversely, limiting dilution assays revealed a

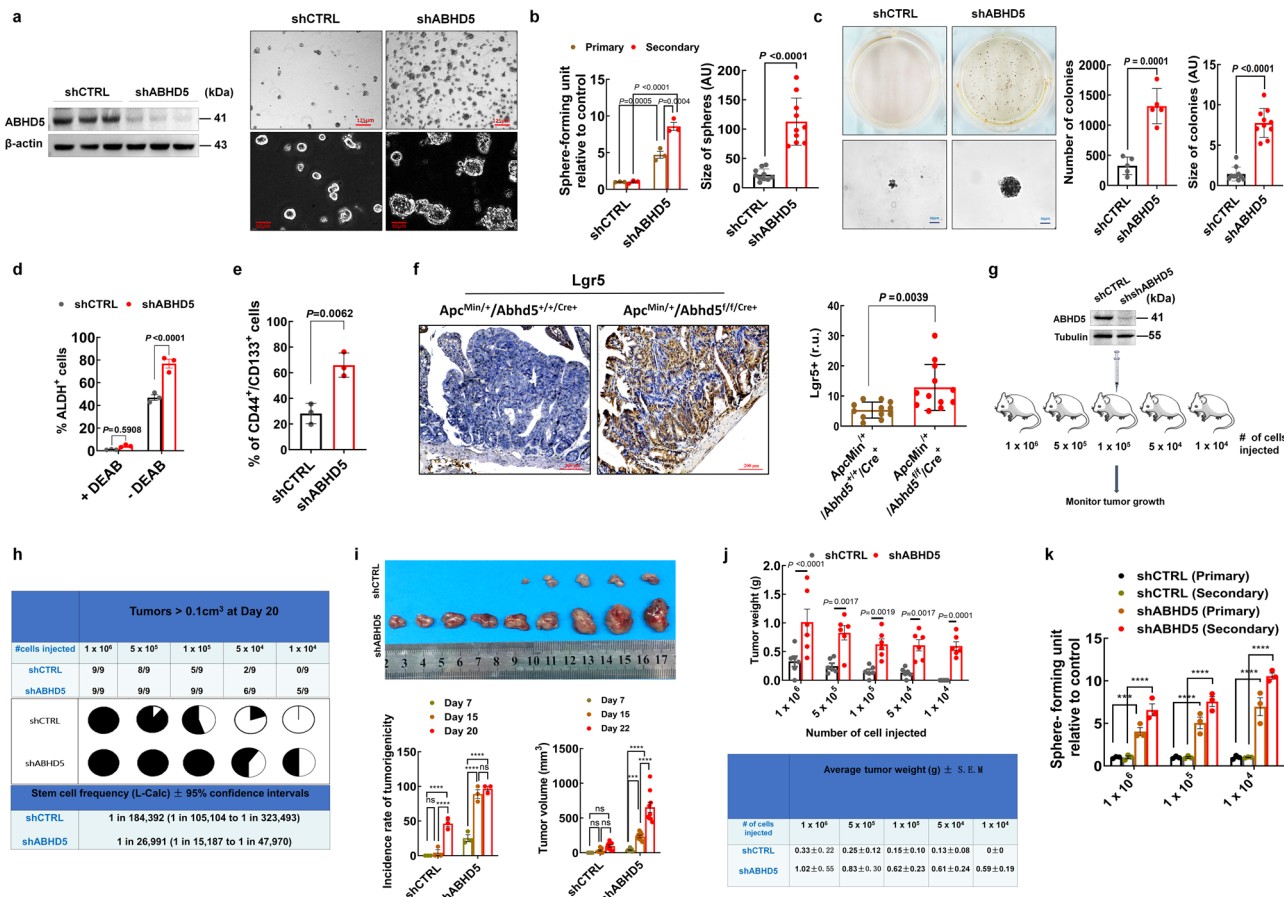

**Fig. 1 Loss of *ABHD5* increases and sustains stemness of CRC cells. a**, **b** Immunoblotting of ABHD5 levels of control (shCTRL) and *ABHD5*-knockdown (shABHD5) HCT116 cells (left). The representative pictures of spheres are shown (right) (**a**). The number and the size ($n = 5$, 10 biologically independent samples) of the spheres were quantified (**b**). AU arbitrary units. **c** The number and the size ($n = 5$, 10 biologically independent samples) of the colonies were quantified. **d** Flow cytometry of ALDH+ proportion in the absence or presence of the ALDH inhibitor diethylaminobenzaldehyde (DEAB; 15 μmol/L) ($n = 3$ biologically independent samples). **e** Flow cytometry of CD133+/CD44+ cells ($n = 3$ biologically independent samples). **f** Representative images of immunohistochemical staining of Lgr5 of intestinal tumour sections from ApcMin/+/Abhd5+/+/Cre+ and ApcMin/+/Abhd5f/f/Cre+ mice. The positive cell number is quantified (right) (r.u. relative units) ($n = 12$ mice/group). **g** Schematic of limiting dilution assay. ABHD5 levels of cells injected were assessed by immunoblotting prior to the experiment (top). **h** Stem cell frequencies were determined using L-Calc software. The upper table shows the number of tumours with a positive response (response = tumour > 0.1 cm³ at 20 days post injection)/total number of tumours and is depicted in the pie chart diagram below. **i** Images of resected xenografts from the group injected with 1×10⁵ shCTRL or sh*ABHD5* HCT116 cells at 15 days after injection. The incidence of tumourigenicity and tumour volume at different time points is quantified (below). ($n = 9$ mice/group, ns no significance ($p > 0.05$), ****$p < 0.0001$). **j** Dot density plot of final tumour weight ($n = 6$ mice/group). **k** The analyses of the relative primary and second sphere-forming unit of shABHD5 or shCTRL HCT116 cells digested from the xenografts ($n = 3$ biologically independent samples, ***$p = 0.0002$, ****$p < 0.0001$) (Data are shown as mean ± s.e.m. Unpaired two-sided Student's *t* test was used in panels (**b**) (right), (**c**), (**e**) and (**f**). Panels (**b**) (left), (**d**) and (**i**–**k**) were analysed using two-way ANOVA and Sidak's multiple comparison test. Source data are provided as a Source Data file.

3.67-fold decrease in stem cell number in *ABHD5*-overexpressing cell xenografts (Supplementary Fig. 1i), and this result was corroborated by the ex vivo sphere formation assay results (Supplementary Fig. 1j). Taken together, these results imply that the ABHD5 inhibits the CRC stem cell population and suppresses the CRC stemness.

**c-Met inhibition is synthetic lethal with loss of ABHD5.** Since *ABHD5* knockdown markedly affects the stemness capacity of CRC cells, we next sought to identify lethal vulnerabilities in *ABHD5*-knockdown CRC cells that may indicate the underlying mechanism. We evaluated 2486 FDA-approved compounds with specific targets for their ability to affect the cell viability of both control and *ABHD5*-knockdown HCT116 cells. We identified 139 compounds that significantly reduced the viability of *ABHD5*-knockdown cells (>90%) with little to no effect on control cells.

Enrichment analysis of the pathways targeted by the identified compounds identified the top categories as angiogenesis, tyrosine kinase/adaptors, DNA damage/repair, chromatin/epigenetics, PI3K/Akt/mTOR signalling, metabolism, and cytoskeletal signalling (Fig. 2a). Inhibitors of c-Met/HGFR ($n = 7$), VEGFR ($n = 7$), and HDAC ($n = 7$) had the greatest representation in the list of identified compounds (Fig. 2b).

As c-Met is a known intrinsic modulator of CSC self-renewal, eliciting a genetic program known as invasive growth[6–11], and the c-Met ligand HGF is secreted by stromal myofibroblasts in colon adenocarcinoma to activate the self-renewal pathway to sustain long-term CSC propagation[23], we chose to focus on the c-Met inhibitors identified in our compound screen. The seven c-Met inhibitors consistently and significantly decreased the viability of *ABHD5*-knockdown HCT116 cells but had modest effects on control cells (Fig. 2c). The drug with the greatest percent

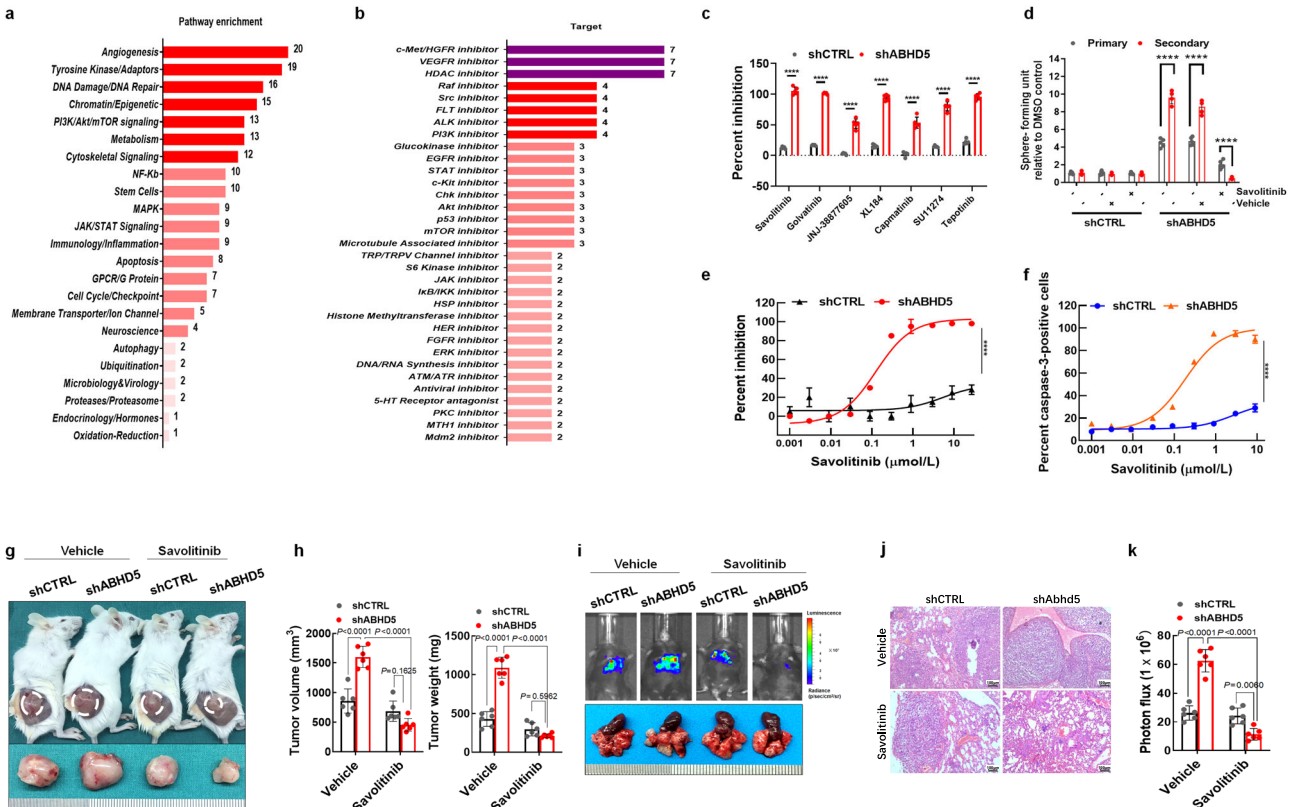

**Fig. 2 c-Met inhibition is synthetic lethal with ABHD5 loss. a**, **b** Pathway enrichment analysis (**a**) and target analysis (**b**) of the FDA-approved compounds that reduced cell viability (>90% viability) of *ABHD5*-knockdown cells (>90%) with little to no effect on control cells. **c** The inhibitory rate analysis of shCTRL or shABHD5 HCT116 cells treated with FDA-approved c-Met inhibitors ($n = 5$ biologically independent samples). **d** Sphere formation assay of HCT116 shCTRL or shABHD5 cells treated with 1 μM savolitinib or vehicle control (DMSO) ($n = 5$ biologically independent samples). **e** Savolitinib dose–response curves in HCT116 shCTRL or shABHD5 cells ($n = 5$ biologically independent samples). **f** Caspase-3 activation in shCTRL or shABHD5 HCT116 cells treated with savolitinib from at least three independent experiments ($n = 5$ biologically independent samples). **g** Subcutaneuos xenografts were established by shCTRL and shABHD5 HCT116 cells in NOD/SCID mice. The tumour-bearing mice were randomized into groups ($n = 6$ mice/group) and dosed by oral gavage as follows: 10 ml/kg vehicle (acidic CMC-Na 0.5%, pH = 2.1) once daily or 2.5 mg/kg savolitinib once daily for 21 days. The representative images of the xenografts were shown. **h** Statistical analyses of the tumour volume and tumour weight in the indicated groups in (**g**) ($n = 6$ mice/group). **i** Lung metastatic xenografts were established by shCTRL and shABHD5 HCT116 cells in NOD/SCID mice. The tumour-bearing mice were randomized into groups ($n = 6$ mice/group) and dosed by oral gavage as follows: 10 ml/kg vehicle (acidic CMC-Na 0.5%, pH = 2.1) once daily or 2.5 mg/kg savolitinib once daily for 21 days. The representative bioluminescent and gross images of lung metastases in NOD/SCID mice were shown. **j** The representative images of HE staining of lung tissues in NOD/SCID mice of (**i**). **k** The quantifications of bioluminescent images of lung metastases in the indicated groups in (**i**) ($n = 6$ mice/group) (Data are shown as mean ± s.e.m. Panels (**c**−**f**), (**h**) and (**k**) were analysed using two-way ANOVA and Sidak's multiple comparison test. ns no significance, ****$p < 0.0001$. Source data are provided as a Source Data file).

inhibition in our screen, the highly potent and selective c-Met inhibitor, savolitinib, significantly suppressed primary and secondary sphere formation by *ABHD5*-knockdown HCT116 cells, with a particularly robust effect on secondary sphere formation (Fig. 2d). Moreover, as shown in Fig. 2e, the dose–response curves revealed that the cytotoxic effects of savolitinib was significantly increased in *ABHD5*-knockdown cells relative to that in control cells, indicating a lethal vulnerabilities of *ABHD5*-knockdown cells to c-Met inhibition[24]. This differential effect presumably results from increased apoptosis, because caspase 3 activation was enhanced in *ABHD5*-knockdown HCT116 cells compared with control cells (Fig. 2f). Importantly, savolitinib robustly decreased the subcutaneous tumour formation and lung metastasis of *ABHD5*-knockdown HCT116 cells in xenograft models (Fig. 2g–k). Collectively, these data show c-Met inhibition is synthetic lethal with *ABHD5* knockdown in CRC cells, strongly suggesting a critical role of c-Met in sustaining the stemness and tumourigenic properties of these cells.

### Loss of ABHD5 activates c-Met to sustain the stemness capacity of CRC cells independent of β-Catenin. As cells with *ABHD5* knockdown were exquisitely sensitive to c-Met inhibition, we next investigated how ABHD5 and c-Met pathways intersect in CRC stemness. *MET* is a well-known target gene of β-catenin, and WNT/β-catenin signalling is known to play a key role in sustaining the stemness of CRC cells. We therefore analysed the expression levels of WNT/β-catenin target genes in *ABHD5*-knockdown HCT116 cells compared with control cells and found significantly higher CyclinD1 (encoded by *CCND1*) and c-Met protein levels in *ABHD5*-knockdown cells (Fig. 3a). Since it has been reported that c-Met overexpression is critically attributable to CRC stemness[10,11], to further confirm the involvement of the increased expression of c-Met in *ABHD5*-knockdown effect on CRC stemness, we silenced *MET* in control and *ABHD5*-knockdown HCT116 cells and analysed stemness characteristics. In these cells, *MET* knockdown abolished the increases in self-renewal and AI growth detected upon *ABHD5* knockdown (Fig. 3b). Similar to the in vitro results, in vivo c-Met expression was higher in intestinal

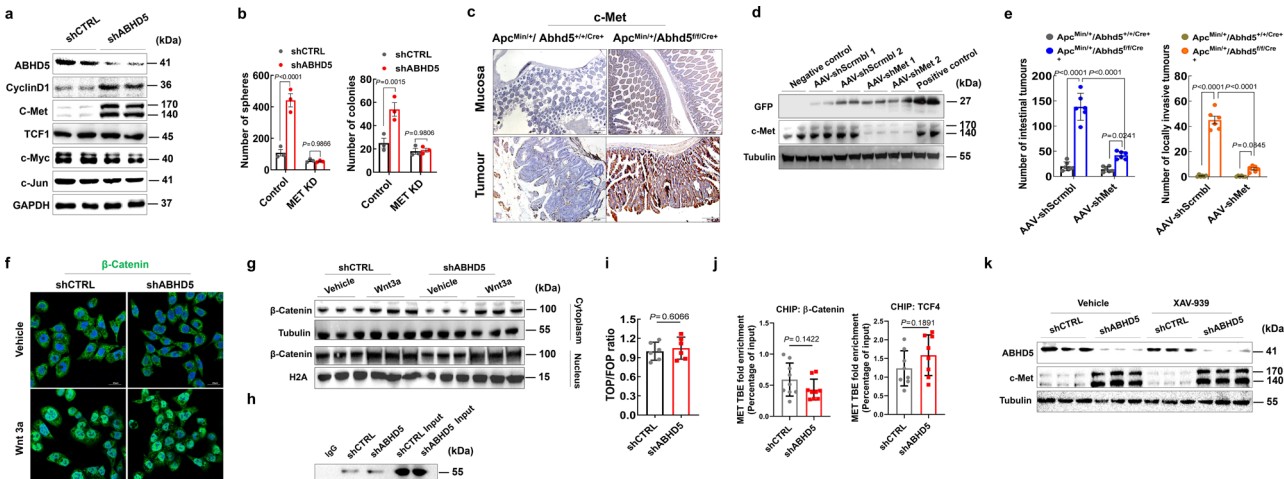

**Fig. 3 Loss of *ABHD5* promotes c-Met transcription to increase and sustain the stemness of CRC cells independent of β-Catenin. a** Immunoblotting of β-Catenin target gene levels in shCTRL or shABHD5 HCT116 cells. **b** Statistical analyses of sphere formation assay and anchorage-independent growth assay of shCTRL or shABHD5 HCT116 cells transfected with either the control shRNA (Control) or *MET*-knockdown shRNA (*MET* KD) ($n = 3$ biologically independent samples). **c** Representative immunohistochemistry images of c-Met in the intestine tumours and intestine mucosa of Apc$^{Min/+}$/Abhd5$^{+/+/Cre+}$ or Apc$^{Min/+}$/Abhd5$^{f/f/Cre+}$ male mice at 100 days of age. Scale bar, 500 μm and 200 μm. **d** Immunoblotting of GFP and c-Met expression levels in the intestinal mucosa of Apc$^{Min/+}$/Abhd5$^{f/f/Cre+}$ mice injected with GFP-labelled control (AAV-shScrmbl) or *Met*-silencing-AAV (AAV-shMet) via the tail vein (iv1–2: tail vein injection in 2 mice). **e** Statistical analysis of the number of total and locally invasive tumours in the entire small intestine of control and Apc$^{Min/+}$/Abhd5$^{f/f/Cre+}$ mice injected with control (AAV-shScrmbl) or *Met*-silencing-AAV (AAV-shMet) via the tail vein ($n = 6$ mice/group). **f** Immunofluorescent staining of β-Catenin in shCTRL or shABHD5 HCT116 treated with either Wnt3a (250 ng/ml) or vehicle (PBS). **g** Subcellular fractionation immunoblotting of cytoplasmic and nuclear β-Catenin in shCTRL or shABHD5 HCT116 cells treated with either Wnt3a (250 ng/ml) or vehicle (PBS). **h** β-Catenin immune complexes were immuno precipitated from shCTRL or shABHD5 HCT116 cells and subjected to immunoblotting of TCF4. **i** Top-flash (Wnt/β-Catenin pathway-responsive firefly luciferase plasmid) reporter gene assay in shCTRL or shABHD5 HCT116 cells ($n = 6$ biologically independent samples). **j** β-Catenin or TCF4 immune complexes were isolated from shCTRL or shABHD5 HCT116 cells and subjected to ChIP-qPCR analysis with primers derived from the promoter regions of *MET* ($n = 9$ (left) or 8 (right) biologically independent samples). **k** Immunoblotting of ABHD5 and c-Met expression levels in shCTRL or shABHD5 HCT116 cells treated with either XAV-939 (10 μM) or vehicle (DMSO) (Data are shown as mean ± s.e.m. Unpaired two-sided Student's *t* test was used in panels (**i**) and (**j**), whereas panels (**b**) and (**e**) were analysed using two-way ANOVA and Sidak's multiple comparison test. Source data are provided as a Source Data file).

tumours and intestinal mucosa of APC$^{Min/+}$ mice, with intestine-specific *Abhd5* knockout (Apc$^{Min/+}$/Abhd5$^{f/f/Cre+}$), than in control mice with intact ABHD5 expression (Fig. 3c). To investigate this further, we specifically knocked down *Met* in the intestine by injecting adeno-associated virus (AAV) expressing shMet under control of the villin promoter (AAV-villin-shMet-GFP, referred to as AAV-shMet) into 6- to 8-week-old Apc$^{Min/+}$/Abhd5$^{f/f/Cre+}$ mice via the tail vein, whereas control mice were injected with AAV-villin-shScrmbl-GFP (AAV-shScrmbl) control virus. As expected, mice infected with AAV-shMet exhibited significantly reduced c-Met levels in intestinal tumours compared with those infected with AAV-shScrmbl (Fig. 3d). In addition, AAV-mediated silencing of *Met* dramatically inhibited the intestinal tumourigenesis and the malignant transformation of intestinal adenomas compared to AAV-shScrmbl in Apc$^{Min/+}$/Abhd5$^{f/f/Cre+}$ mice (Fig. 3e). This evidence confirms that c-Met is critically involved in the mechanism by which loss of *ABHD5* increases and sustains the stemness of CRC cells, and these results provide an explanation for the sensitivity of *ABHD5*-knockdown cells to c-Met inhibitors.

Given the involvement of c-Met in CRC cell stemness and tumourigenesis in the context of loss of *ABHD5*, we next sought to identify the mechanism by which ABHD5 regulates c-Met. We first examined whether ABHD5 regulates c-Met in a β-Catenin-dependent manner. As measured by immunofluorescence and fractionation assays, the β-Catenin levels and localization did not differ between *ABHD5*-knockdown and control HCT116 cells under the stimulation of Wnt3a, a potent factor inducing the nuclear translocation of β-Catenin (Fig. 3f, g and Supplementary Fig. 2a). Moreover, β-Catenin expression and localization in intestinal tumours and intestinal mucosa did not differ between

Apc$^{Min/+}$/Abhd5$^{f/f/Cre+}$ and control mice (Supplementary Fig. 2b, c). Since ABHD5 plays little effect on the expression and nuclear translocation of β-Catenin, we evaluated whether ABHD5 affects the β-Catenin/TCF interaction by immunoprecipitation of nuclear *ABHD5*-knockdown and control HCT116 cell lysates with an anti-β-Catenin antibody or mouse IgG (control) and western blotting for TCF4, and found that loss of *ABHD5* did not affect the β-Catenin/TCF4 interaction (Fig. 3h). Correspondingly, there was no difference in TOPFlash reporter activity between *ABHD5*-knockdown and control HCT116 cells (Fig. 3i), indicating that ABHD5 does not alter β-Catenin stability. Consistent with this result, chromatin immunoprecipitation (ChIP)-qPCR assays using anti-β-Catenin and anti-TCF4 antibodies showed that *ABHD5* knockdown did not increase the binding of either β-Catenin or TCF4 to the *MET* promoter region (Fig. 3j). As further confirmation that β-Catenin is not involved in altering c-Met levels in this context, treatment with the β-Catenin inhibitor XAV939 did not affect c-Met levels in *ABHD5*-knockdown cells (Fig. 3k and Supplementary Fig. 2d). In the last set of experiments confirming the lack of β-Catenin involvement, we used the RKO cell line, which is unique among CRC cell lines in not showing active β-Catenin/TCF-regulated transcription. In accordance with our previous results, RKO cells showed increased c-Met expression upon *ABHD5* knockdown, despite the lack of β-Catenin transcriptional activity (Supplementary Fig. 3a), and silencing *MET* in *ABHD5*-knockdown RKO cells significantly suppressed self-renewal and AI growth (Supplementary Fig. 3b, c). Collectively, these findings strongly suggest that upon the loss of *ABHD5*, c-Met is regulated via a pathway independent of β-Catenin.

**Loss of ABHD5 activates c-Met transcription by promoting YAP nuclear localization and activation**. Given the lack of β-Catenin involvement in the interrogated pathway, we searched for other factors involved in regulating c-Met expression independent of β-Catenin. YAP is an alternative transcriptional coactivator required for the activation of β-Catenin target genes in multiple cancers[25–28]. Interestingly, YAP and β-Catenin are recruited to and activate common target genes (including *CCND1* and *MET*) with the transcription factors TEAD and TCF, respectively[29]. Moreover, YAP function is reported to be required for intestinal tumourigenesis resulting from APC loss[30,31]. Therefore, we sought to determine whether ABHD5 deficiency promotes c-Met transcription and CRC pathogenesis in a YAP-dependent manner. We examined the expression patterns of β-catenin-target and YAP-target genes in *ABHD5*-knockdown and control HCT116 cells by Microarray. Notably, YAP target genes but not β-Catenin target genes showed a significant shift between *ABHD5*-knockdown and control HCT116 cells (Fig. 4a). The KEGG (Kyoto Encyclopedia of Genes and Genomes) pathway analysis further showed that the Hippo signalling pathway, of which YAP is a downstream effector, and signalling pathways regulating stem cell pluripotency were among the top 20 enriched pathways in *ABHD5*-knockdown HCT116 cells relative to control HCT116 cells (Fig. 4b).

The above results suggested that YAP signalling was indeed activated in cells with low ABHD5 expression; to corroborate this, we measured YAP nuclear accumulation in *ABHD5*-knockdown HCT116 cells and found that the levels were higher compared to those in control cells (Fig. 4c, d). Accordingly, co-immunoprecipitation assays showed a higher level of the transcription factor TEAD interacting with YAP (Fig. 4e). These increases were accompanied by transcriptional activation of both a synthetic TEAD reporter (8xGTIIC-Lux) (Fig. 4f) and the direct YAP target genes *Gh2*, *CTGF*, *DKK1*, *ITGB2*, *Birc5* and *AREG* (Fig. 4g). Treatment with verteporfin, a potent inhibitor of the interaction between YAP and TEAD, or knockdown of *YAP* strongly reversed the increase in c-Met expression in *ABHD5*-knockdown HCT116 cells (Fig. 4h), indicating that the increased c-Met levels result from transactivation by YAP.

To confirm the involvement of YAP in *ABHD5*-knockdown downstream cellular consequences, we treated HCT116 cells and xenograft mouse models with verteporfin or vehicle and found that verteporfin significantly abolished the increased self-renewal and AI growth of *ABHD5*-knockdown cells in vitro (Supplementary Fig. 4a, b) and inhibited the growth of *ABHD5*-knockdown cell xenografts in vivo (Supplementary Fig. 4c). These findings were confirmed in murine CRC cells MC-38 and CT-26 (Supplementary Fig. 5), suggesting that modulation of YAP- c-Met signalling by ABHD5 is conserved between murine and human CRCs.

To verify the involvement of YAP in intestinal tumourigenesis influenced by ABHD5, we utilized the Apc$^{Min/+}$/Abhd5$^{f/f/Cre+}$ model. Strikingly, the nuclear translocation of YAP was dramatically increased in an *ABHD5* gene dosage-dependent manner in intestinal tumours of Apc$^{Min/+}$/Abhd5$^{f/+/Cre+}$ and Apc$^{Min/+}$/Abhd5$^{f/f/Cre+}$ mice relative to control (Apc$^{Min/+}$/Abhd5$^{+/+/Cre+}$) mice (Fig. 4i). In addition, c-Met levels were significantly lower in intestinal tumours of Apc$^{Min/+}$/Abhd5$^{f/f/Cre+}$ mice infected with AAV-shYap than in those of AAV-shScrmbl-infected mice (Fig. 4j). Additionally, AAV-induced *Yap* silencing dramatically inhibited intestinal tumour formation and malignant transformation in Apc$^{Min/+}$/Abhd5$^{f/f/Cre+}$ mice (Fig. 4k). Collectively, the evidence identified YAP as a critical effector mediating ABHD5-regulated c-Met expression and CRC stemness.

**ABHD5 deficiency promotes nuclear localization and activation of YAP via inducing SET1A-mediated YAP methylation at K342**. We next aimed to further elucidate the mechanism underlying the nuclear localization and activation of YAP in the context of ABHD5 deficiency. First, we measured whether ABHD5 affected YAP expression, however, neither the mRNA nor the total protein level of YAP differed between control and *ABHD5*-knockdown cells (Fig. 5a and Supplementary Fig. 6a). As several studies have shown that the localization and activity of important tumour suppressors/oncogenes are regulated by post-translational modifications[32–35] and evidence has emerged that phosphorylation or methylation of YAP controls YAP localization and activation[33], we investigated whether ABHD5 regulates YAP activity by modulating its post-translational modification. Specifically, we measured the phosphorylation and methylation status of YAP in HCT116 cells by mass spectrometry and found decreased phosphorylation at S127 and increased methylation at K342, an evolutionarily conserved lysine residue, in *ABHD5*-knockdown cells compared to control cells (Fig. 5b). It has been demonstrated that K342 methylation of YAP sequesters YAP in the nucleus, and prevents the translocation of YAP to the cytoplasm for phosphorylation and ubiquitination/degradation[33]. We speculate that the decreased phosphorylation of YAP at S127 is a consequence of increased methylation of YAP at K342. To confirm that YAP K342 is mono-methylated under *ABHD5*-knockdown conditions, we analysed cell lysates with an antibody specific to K342-mono-methylated YAP[33] and found significantly higher levels of K342-methylated YAP in *ABHD5*-knockdown cells than in control cells (Fig. 5c and Supplementary Fig. 6b). Similar results were also observed in SW620 CRC cells (Supplementary Fig. 6c). Moreover, mutation of K342 substantially decreased YAP methylation at this residue in the context of *ABHD5* knockdown (Fig. 5d). To confirm the effect of YAP methylation on its phosphorylation and activity, we treated HCT116 cells with the demethylase LSD1, which dose-dependently reversed YAP methylation at K342 and phosphorylation at S127 in *ABHD5*-knockdown cells (Fig. 5e), and abolished YAP-mediated TEAD activity in *ABHD5*-knockdown cells (Fig. 5f). Given that YAP methylation at K342 blocks its CRM1-mediated nuclear export[33], hereby enabling sustained YAP transcriptional activity, we evaluated the interaction between YAP and CRM1 in co-IP assays and showed that *ABHD5* knockdown decreased this interaction (Fig. 5g). Collectively, these data suggest that *ABHD5* knockdown leads to increased YAP methylation at K342 to promote YAP nuclear localization and activity.

To further delineate the mechanism involving YAP in CRC with loss of *ABHD5*, we first confirmed that YAP methylation at K342 is catalysed by SET1A[33]. We individually knocked down a panel of lysine methyltransferases and found that *SET1A* knockdown had the greatest effect on YAP methylation in *ABHD5*-knockdown HCT116 cells (Fig. 5h). Moreover, *SET1A* knockdown decreased YAP nuclear localization (Fig. 5i, j), TEAD activity (Fig. 5k), and c-Met levels (Fig. 5l). Furthermore, co-IP assays of nuclear protein confirmed the interaction between SET1A and YAP, which was increased in *ABHD5*-knockdown cells compared with control cells (Fig. 5m). Collectively, the evidence presented here suggests that SET1A methylation of YAP is significantly increased under *ABHD5* knockdown conditions and is key for the K342 methylation, nuclear localization, and transactivation of *YAP*.

**Loss of ABHD5 increases SET1A-mediated histone methylation to increase chromatin accessibility and promote YAP-induced c-Met transcription**. SET1A is a member of the mammalian SET1/MLL family of H3K4MTs, which also catalysers

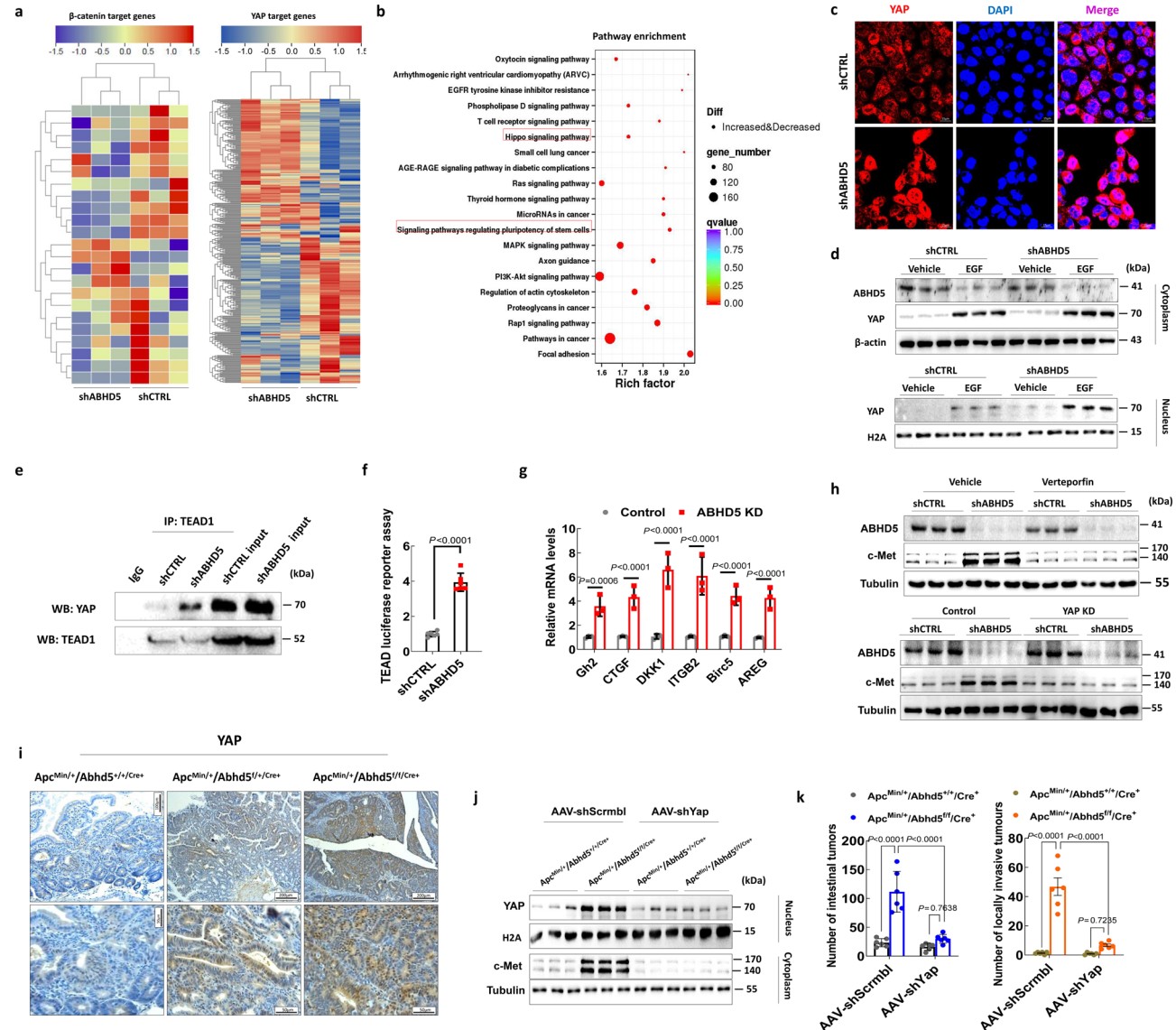

**Fig. 4 Loss of ABHD5 promotes c-Met transcription via promoting the nuclear localization and activation of YAP. a** Heatmap showing the expression patterns of β-Catenin or YAP target genes in shCTRL or shABHD5 HCT116 cells screened by Microarray. **b** KEGG pathway enrichment analysis of differentially expressed genes between shCTRL or shABHD5 HCT116 cells screened by Microarray. The top 20 positively and negatively enriched pathways are shown in the bar plot. **c** Representative immunofluorescent staining of YAP in shCTRL or shABHD5 HCT116 cells. **d** Subcellular fractionation immunoblotting of cytosolic and nucleic YAP in shCTRL or shABHD5 HCT116 cells treated with either EGF (200 ng/ml) or vehicle (PBS). **e** TEAD immune complexes were immunoprecipitated from shCTRL or shABHD5 HCT116 cells and subjected to immunoblotting of YAP. **f** TEAD luciferase reporter assay in shCTRL or shABHD5 HCT116 cells ($n = 3$ biologically independent samples). **g** Real-time PCR assay of the indicated target gene mRNA levels in shCTRL or shABHD5 HCT116 cells ($n = 3$ biologically independent samples). **h** Immunoblotting of ABHD5 and c-Met expression levels in shCTRL or shABHD5 HCT116 cells treated with YAP knockdown or YAP inhibitor verteporfin (15 μM), DMSO as a vehicle control. **i** Representative immunohistochemistry staining of YAP expression in paraffin sections of the intestinal tumours of 100-day-old male mice. $Apc^{Min/+}/Abhd5^{+/+}/Cre+$, $Apc^{Min/+}/Abhd5^{f/+}/Cre+$ and $Apc^{Min/+}/Abhd5^{f/f}/Cre+$. Scale bar, 200 μm and 50 μm. **j** Immunoblotting of YAP and c-Met expression levels in the intestinal tumours of $Apc^{Min/+}/Abhd5^{+/+}/Cre+$ and $Apc^{Min/+}/Abhd5^{f/f}/Cre+$ mice injected with AAV-Scrmbl or AAV-shYap via the tail vein. H2A as a loading control for nuclear protein, and tubulin as a loading control for cytoplasmic protein. **k** Statistical analysis of the number of total and locally invasive tumours in the entire small intestine of $Apc^{Min/+}/Abhd5^{+/+}/Cre+$ and $Apc^{Min/+}/Abhd5^{f/f}/Cre+$ mice injected with AAV-Scrmbl or AAV-shYap via the tail vein ($n = 6$ mice/group) (Data are shown as mean ± s.e.m. Unpaired two-sided Student's $t$ test was used in panel (**f**), whereas panels (**g**) and (**k**) were analysed using two-way ANOVA and Sidak's multiple comparison test. Source data are provided as a Source Data file).

H3K4 mono-, di- and trimethylation to regulate gene expression[35]. Since we demonstrated that the activity of SET1A was regulated by ABHD5, we cannot exclude that SET1A-mediated histone methylation may also contribute to ABHD5 deficiency-induced transcription of c-Met. Expectedly, methylated H3K4 (H3K4me) levels were substantially higher in *ABHD5*-knockdown HCT116 cells relative to control cells (Supplementary Fig. 7a), and the assay for targeting accessible-chromatin with high-throughput sequencing (ATAC-seq) revealed a shift of chromatin accessibility in *ABHD5*-knockdown cells relative to that in control cells (Supplementary Fig. 7b), suggesting that SET1A-mediated histone methylation and chromatin remodelling may play a synergistic effect on c-Met transcription. We thus completed multiomic profiling of

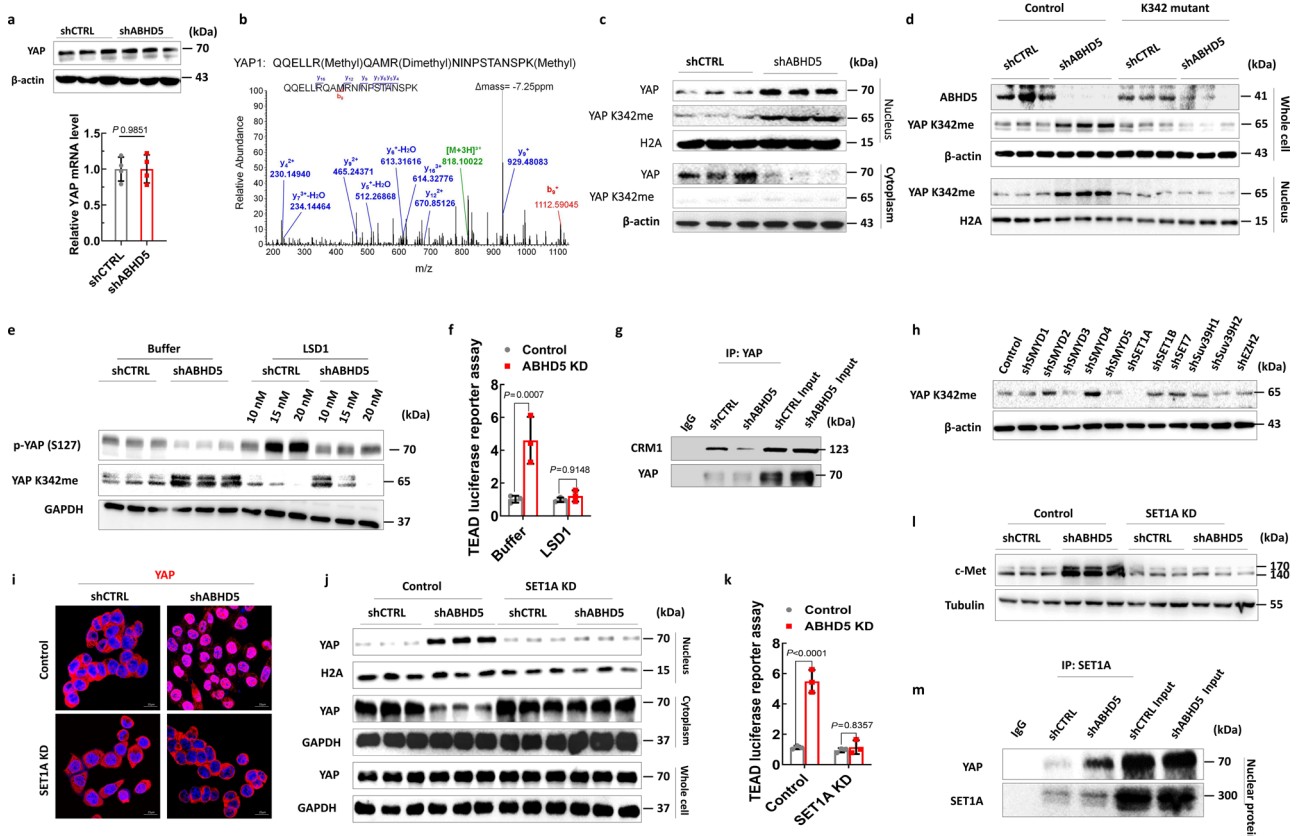

**Fig. 5 ABHD5 deficiency promotes YAP activity and nuclear localization via inducing SET1A-mediated YAP methylation. a** Real-time PCR and immunoblotting analyses of YAP expression levels in shCTRL or shABHD5 HCT116 cells ($n = 3$ biologically independent samples). **b** Tandem mass spectrometry spectrum of Glu-C-digested YAP 332–349 fragment (methylated K342). Detected productions are indicated in red (b ions) and blue (y ions). **c** Subcellular factionation immunoblotting of YAP and methylated YAP at k342 (YAP K342me) in shCTRL or shABHD5 HCT116 cells. **d** Immunoblotting of YAP K342m expression levels in shCTRL or shABHD5 HCT116 cells transfected with control vector (pcDNA3.1) or YAP 342 mutant (K342 mutant). **e** Immunoblotting of phosphorylated YAP (p-YAP S127) and YAP K342m expression levels in shCTRL or shABHD5 HCT116 cell homogenates treated with control buffer or LSD1. **f** TEAD luciferase reporter assay in shCTRL or shABHD5 HCT116 cell homogenates treated with control buffer or LSD1 (20 nM) ($n = 3$ biologically independent samples). **g** YAP immune complexes were immunoprecipitated from shCTRL or shABHD5 HCT116 cells and subjected to immunoblotting of YAP and CRM1. **h** Immunoblotting of YAP K342m expression levels in shABHD5 HCT116 cells transfected with control or shRNA plasmids silencing the indicated histone methyltransferases. **i** Representative immunofluorescent staining of YAP in shCTRL or shABHD5 HCT116 cells transfected with control or *SET1A*-shRNA plasmids (SET1A KD). **j** Immunoblotting of nucleic YAP expression levels in shCTRL or shABHD5 HCT116 cells transfected with control or SET1A-shRNA plasmids (SET1A KD). **k** TEAD luciferase reporter assay in shCTRL or shABHD5 HCT116 cells transfected with control or *SET1A*-shRNA plasmids (SET1A KD) ($n = 3$ biologically independent samples). **l** Immunoblotting of c-Met expression levels in shCTRL or shABHD5 HCT116 cells transfected with control or *SET1A*-shRNA plasmids (SET1A KD). **m** SET1A immune complexes were immunoprecipitated from shCTRL or shABHD5 HCT116 cells and subjected to immunoblotting of YAP (Data are shown as mean ± s.e.m. Unpaired two-sided Student's *t* test was used in panel (**a**), and panels (**f**) and (**k**) were analysed using two-way ANOVA and Sidak's multiple comparison test. Source data are provided as a Source Data file).

*ABHD5*-knockdown and control HCT116 cells to extensively map (1) chromatin conformation (Hi-C); (2) histone methylation (H3K4m3) (ChIP-seq); (3) chromatin accessibility (ATAC-seq) and (4) transcriptomic (RNA-seq) signatures surrounding the *MET* locus on chromosome 5. As shown in Supplementary Fig. 7c, an increased H3K4me3 level, chromatin accessibility and SET1A binding, accompanied by increased c-Met transcription were observed in *ABHD5*-knockdown cells compared with control cells. Collectively, these data strongly support the hypothesis that loss of *ABHD5* also promotes SET1A-induced histone methylation, which increases chromatin accessibility and synergistically promotes YAP-mediated c-Met transcription.

**ABHD5 impedes SET1A-induced YAP methylation by a noncanonical mechanism involving sequestering DPY30 in the cytoplasm for ubiquitination.** Given the above finding that loss of *ABHD5* leads to increased SET1A methylation of YAP, we next

wanted to ascertain the underlying mechanism. Since ABHD5 acts as a cofactor of patatin-like phospholipase domain-containing 2 (PNPLA2, also known as ATGL), as an initial step in investigating the underlying mechanism, we aimed to determine whether PNPLA2, or the loss of the interaction between PNPLA2 and ABHD5, is involved in YAP activation in the context of *ABHD5* knockdown. PNPLA2 levels were substantially increased in *ABHD5*-knockdown HCT116 cells (Supplementary Fig. 8a). Given that PNPLA2 is expressed in HCT116 cells and is upregulated upon *ABHD5* knockdown, we next knocked down *PNPLA2* in HCT116 cells and evaluated protein levels and localization. Intriguingly, *PNPLA2* knockdown in HCT116 cells did not affect c-Met expression but did significantly increase the levels of c-Jun, CyclinD1 and CD44, the important β-Catenin target genes for tumourigenesis (Supplementary Fig. 8b). Meanwhile, the expression and nuclear localization of β-Catenin were significantly increased in *PNPLA2*-knockdown HCT116 cells

compared with control cells (Supplementary Fig. 8c, d). Moreover, co-IP assays showed an increase in the β-Catenin/TCF4 interaction and a decrease in the YAP/TEAD1 interaction in *PNPLA2*-knockdown HCT116 cells compared with control cells (Supplementary Fig. 8e). Also, the TOPFlash and TEAD luciferase reporter assays confirmed that transcriptional activity downstream of β-Catenin, but not YAP, was activated in *PNPLA2*-knockdown HCT116 cells (Supplementary Fig. 8f, g). Importantly, modulation of *ABHD5* knockdown in *PNPLA2*-null HCT116 cells significantly increased YAP methylation at K342 and activation (Supplementary Fig. 8h, i). The evidence seems to show that PNPLA2 is not involved in ABHD5-regulated YAP methylation and c-Met expression. To further confirm this deduction, we used two *ABHD5* mutants[36]. As shown in Supplementary Fig. 8j, k, the introduction of either *ABHD5* Q130P or E260K into *ABHD5*-null HCT116 cells successfully abolished YAP methylation and activation, suggesting that ABHD5 negatively regulates YAP activation in a manner functionally independent of its role as a coactivator of PNPLA2.

In addition to activating PNPLA2, ABHD5 catalysers the acylation of lysophosphatidic acid (LPA) to generate the critical lipid second messenger phosphatidic acid (PA)[37]. Given that LPA was reported to contribute to YAP dephosphorylation and nuclear translocation[38], we examined the involvement of LPA in YAP methylation and nuclear localization under conditions of *ABHD5* knockdown. Intriguingly, although knocking down *ABHD5* in HCT116 cells dramatically increased LPA secretion and LPA receptor 1 (LPAR1) expression (Supplementary Fig. 9a, b), the LPAR1 inhibitor Ki16425 [39] only modestly affected YAP methylation at K342 and activation (Supplementary Fig. 9c, d). In addition, Ki16425 failed to abolish the increases in c-Met levels (Supplementary Fig. 9e), sphere formation and AI growth of *ABHD5*-knockdown HCT116 cells (Supplementary Fig. 9f), indicating that LPA signalling is not the predominant mechanism underlying enhanced YAP methylation and activation in the context of *ABHD5* deficiency.

Since the canonical signalling pathways excluded, we deduced that ABHD5 might regulate SET1A methylation of YAP via a non-canonical signalling pathway. SET1A complex possesses an evolutionarily conserved subcomplex called WRAD, which is composed of WDR5, RbBP5, ASH2L, and DPY30 and is required for SET1A methyltransferase activity[40]. To investigate the mechanism underlying YAP methylation upon *ABHD5* depletion, we introduced shRNA targeting *RbBP5*, *ASH2L*, *WDR5*, or *DPY30* in *ABHD5*-knockdown and control HCT116 cells. Only knockdown of *DPY30*, the core subunit of the SET1A complex, substantially decreased the increase in YAP methylation in *ABHD5*-knockdown HCT116 cells (Fig. 6a). As a core member of the SET/MLL family, DPY30 directly modulates H3K4 methylation by MLL family complexes, and loss of DPY30 significantly reduces H3K4 methylation[41,42]. DPY30 has also emerged as an important factor in the regulation of hematopoietic progenitor cell proliferation and differentiation and been implicated as an oncogene[43]. We then examined the potential relevance of DPY30 in ABHD5-regulated YAP methylation and c-Met expressions. Total DPY30 protein levels were significantly increased in *ABHD5*-knockdown HCT116 cells compared with control cells (Fig. 6b and Supplementary Fig. 10a), and silencing *DPY30* in *ABHD5*-knockdown HCT116 cells reversed the increases in c-Met levels (Fig. 6c) and YAP transcriptional activity (Fig. 6d). Expectedly, silencing *ABHD5* in SW620 also increased the protein expression levels of DPY30 (Supplementary Fig. 10b). In accordance with these findings, *DPY30* silencing in *ABHD5*-knockdown HCT116 cells significantly decreased self-renewal and AI growth in vitro (Supplementary Fig. 10c) and inhibited xenograft tumour growth and metastasis in vivo (Supplementary Fig. 10d). Additionally, the histopathological analysis showed that DPY30 expression levels were substantially higher in intestinal tumours from Apc$^{Min/+}$/Abhd5$^{f/f/Cre+}$ mice than in those from control Apc$^{Min/+}$/Abhd5$^{+/+/Cre+}$ mice (Fig. 6e). In addition, AAV-induced *Dpy30* silencing in Apc$^{Min/+}$/Abhd5$^{f/f/Cre+}$ mice abolished the increases in YAP methylation and c-Met levels in intestinal tumours (Fig. 6f) and significantly inhibited tumour formation and malignant transformation (Fig. 6g). These findings indicate a critical role for DPY30 in mediating the effect of *ABHD5* knockdown on SET1A-induced YAP methylation and c-Met expression.

We next sought to determine the mechanism by which ABHD5 regulates DPY30. Although DPY30 protein levels were significantly increased in *ABHD5*-knockdown HCT116 cells compared to control cells, *DPY30* mRNA levels were not affected (Fig. 6h). Therefore, we studied other aspects of DPY30 protein regulation, including stability, localization, and post-translational modification. ABHD5 is a cytoplasmic protein, and DPY30 has been shown to shuttle between the cytoplasm and nucleus[44]. Interestingly, the nuclear localization of DPY30 was markedly increased upon *ABHD5* knockdown (Fig. 6i and Supplementary Fig. 10e), leading us to speculate that ABHD5 interacts with DPY30 to retain it in the cytoplasm and thus block it from interacting with YAP and activating target gene *MET* transcription. Immunofluorescence (Fig. 6j) and co-IP assays (Fig. 6k) indicated an interaction between ABHD5 and DPY30 in control HCT116 cells, whereas *ABHD5*-knockdown cells showed marked increases in the nuclear colocalization of and interaction between DPY30 and YAP. To further examine the effect of ABHD5-DPY30 interaction on DPY30 activity, we compared the DPY30 antibody-immunoprecipitated protein complexes between control and *ABHD5*-knockdown HCT116 cells. Based on the differentially expressed proteins between *ABHD5*-knockdown and control HCT116 cells, GO enrichment analysis confirmed that the DPY30-associated proteins in *ABHD5*-knockdown HCT116 cells are mostly enriched in nucleus, and the histone methylation was the top shift molecular function in *ABHD5*-knockdown HCT116 cells relative to control HCT116 cells (Supplementary Fig. 11a, b). The evidence demonstrated that *ABHD5* interacts with DPY30 to prevent it from translocating into the nucleus. Importantly, protein−protein docking showed a "key-lock hole" interaction mode between ABHD5 and DPY30 (Fig. 6l). We further probed a human proteome microarray (HPM) consisting of 20,240 individual N-terminally glutathione S-transferase (GST)-tagged proteins with biotinylated ABHD5 protein. Briefly, recombinant human ABHD5 protein was biotinylated, and the resultant ABHD5−biotin conjugates were incubated with the HPM, and proteins with ABHD5-binding capacity were identified by adding Cy5-conjugated streptavidin. Using the stringent criteria as described in the "Methods" section, 271 proteins were identified as potential ABHD5-interacting proteins, including DPY30 (Fig. 6m). Collectively, these data indicate that ABHD5 interacts with DPY30 to prevent its nuclear translocation, which would impair SET1A activity and SET1A-mediated YAP methylation in the nucleus; in the absence of ABHD5, DPY30 is released and translocate to the nucleus, where it interacts with YAP to support c-Met transcription and downstream cellular consequences.

Next, we investigated the mechanisms by which ABHD5 regulates DPY30 in CRC cells. Knocking down *ABHD5* had no effects on *DPY30* mRNA expression (Fig. 6h), implying that the regulation is not at the transcriptional level and may be at the post-translational level. Since DPY30 protein levels increased upon *ABHD5* knockdown and ABHD5 interacts with DPY30 in cytoplasm, we speculated that ABHD5 might retain DPY30 in the cytoplasm for degradation. Expectedly, western blot analyses of

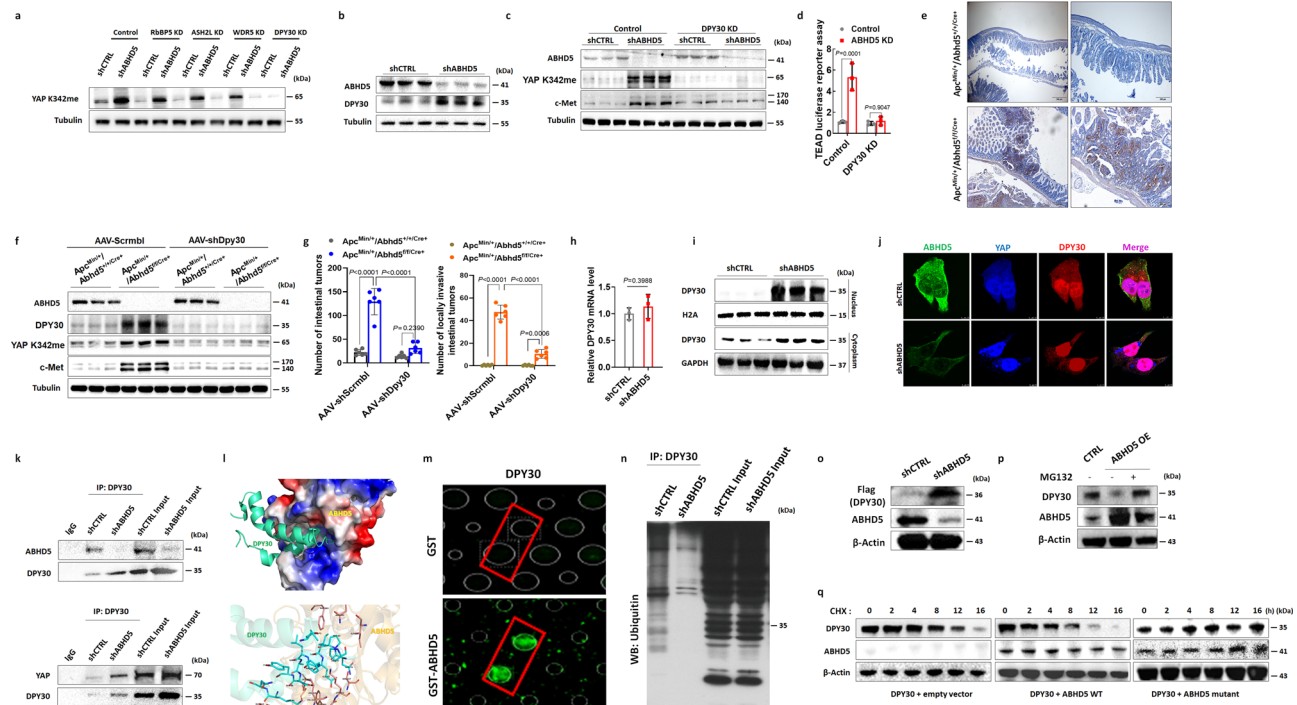

**Fig. 6 ABHD5 impedes SET1A activity via sequestering DPY30, the core unit of SET1A complex, in cytoplasm. a** Immunoblotting of YAP K342me levels in shCTRL or shABHD5 HCT116 cells transfected with control or siRNA plasmids silencing the indicated SET1A subunits. **b** Immunoblotting of DPY30. **c** Immunoblotting of ABHD5, YAP K342me and c-Met in shCTRL and shABHD5 HCT116 cells transfected with control vector or *DPY30* shRNA plasmids (*DPY30* KD). **d** TEAD luciferase reporter assay in shCTRL and shABHD5 HCT116 cells transfected with control vector or *DPY30* shRNA plasmids (*DPY30* KD) ($n = 3$ biologically independent samples). **e** Representative immunohistochemistry staining of DPY30 in the intestinal tumours of Apc$^{Min/+}$/Abhd5$^{+/+/Cre+}$ and Apc$^{Min/+}$/Abhd5$^{f/f/Cre+}$. **f** Immunoblotting of YAP K342me and c-Met in the intestinal tumours of Apc$^{Min/+}$/Abhd5$^{+/+/Cre+}$ and Apc$^{Min/+}$/Abhd5$^{f/f/Cre+}$ mice injected with AAV-Scrmbl or AAV-shDpy30 via tail vein. **g** Statistical analysis of intestinal tumours in Apc$^{Min/+}$/Abhd5$^{+/+/Cre+}$ and Apc$^{Min/+}$/Abhd5$^{f/f/Cre+}$ mice injected with AAV-Scrmbl or AAV-shDpy30 ($n = 6$ mice/group). **h** Real-time PCR assay of *DPY30* mRNA levels ($n = 3$ biologically independent samples). **i** Immunoblotting of cytoplasmic and nuclear DPY30. **j** Representative immunofluorescent staining of ABHD5, YAP and DPY30. **k** DPY30 immune complexes were immunoprecipitated from shCTRL or shABHD5 HCT116 cells and subjected to immunoblotting of ABHD5, YAP and DPY30. **l** The interaction complex model between ABHD5 and DPY30 predicted by docking methods. **m** A direct interaction between GST-labelled-ABHD5 and DPY30 based on Human Proteome Microarray assay. **n** Ubiquitin levels of DPY30 immune complexes. **o** Immunoblotting of Flag-DPY30 expression using Flag antibody in *ABHD5*-knockdown HCT116 cells transfected with Flag-*DPY30*. **p** Immunoblotting of the indicated proteins in vector control or *ABHD5*-overexpressing HCT116 cells in the presence of MG132 (5 mM) for 16 h. **q** Immunoblotting showing the stability of DPY30 protein in HCT116 cells transfected with hemagglutinin-tagged DPY30 (HA-*DPY30*) and WT Flag-*ABHD5* or mutant Flag- *ABHD5* deleted the binding domain to DPY30. Cells were treated with 20 mM cycloheximide (CHX) at the indicated intervals (Data are shown as mean ± s.e.m. Unpaired two-sided Student's *t* test was used in panel (**h**), whereas panels (**d**) and (**g**) were analysed using two-way ANOVA and Sidak's multiple comparison test. Source data are provided as a Source Data file).

proteins immunoprecipitated with the anti-DPY30 antibody showed lower ubiquitin levels in *ABHD5*-knockdown cells than in control cells (Fig. 6n). Indeed similar to the endogenous DPY30, exogenous expression of Flag-tagged *DPY30* was also elevated in *ABHD5*-knockdown HCT116 cells (Fig. 6o). Consistently, the addition of proteasome inhibitor MG132 abrogated ABHD5-mediated DPY30 downregulation in HCT116 cells (Fig. 6p, lanes 3 vs. 2), suggesting that ABHD5 downregulates DPY30 protein expression in a proteasome-dependent manner. Pulse-chase analysis using cycloheximide indicated that over-expression of WT but not mutant *ABHD5* (DPY30 binding domain mutated) reduced the DPY30 protein half-life in HCT116 cells (Fig. 6q), suggesting that ABHD5-mediated DPY30 down-regulation requires the interaction between ABHD5 and DPY30. Taken together, these data support a model wherein ABHD5 sequesters DPY30 in the cytoplasm for ubiquitination and proteasome-dependent degradation, and thus, ABHD5 deficiency increases the nuclear translocation of DPY30 to achieve SET1A activity and SET1A-induced YAP methylation.

**Pharmacological inhibition of c-Met reduces the stemness of primary CRC cells derived from ABHD5$^{low}$DPY30$^{high}$ c-Met$^{high}$ tumours.** Since the above findings strongly suggest the critical role of ABHD5-DPY30- c-Met signalling in regulating the stemness of CRCs, we want to ascertain their potential translation into the clinic. We conducted immunohistochemistry (IHC) of CRC tissue microarrays with available patient outcome data to evaluate ABHD5, DPY30, and c-Met expression. Based on the immunostaining scores of ABHD5, DPY30, and c-Met, these tissues were divided into the high (scores of 2–3) and low groups (scores of 0–1). High ABHD5 expression significantly correlated with low DPY30 and c-Met expression, low ABHD5 expression significantly correlated with high DPY30 and c-Met expression, and there was also a positive correlation between DPY30 and c-Met (Fig. 7a, b). In the outcome analysis, the patients with ABHD5$^{low}$DPY30$^{high}$ c-Met$^{high}$ tumours had poorer disease-free survival (DFS) than those with ABHD5$^{high}$DPY30$^{low}$ c-Met$^{low}$ tumours (Fig. 7c), suggesting the potential value of ABHD5-DPY30- c-Met expression in predicting the prognosis of CRCs.

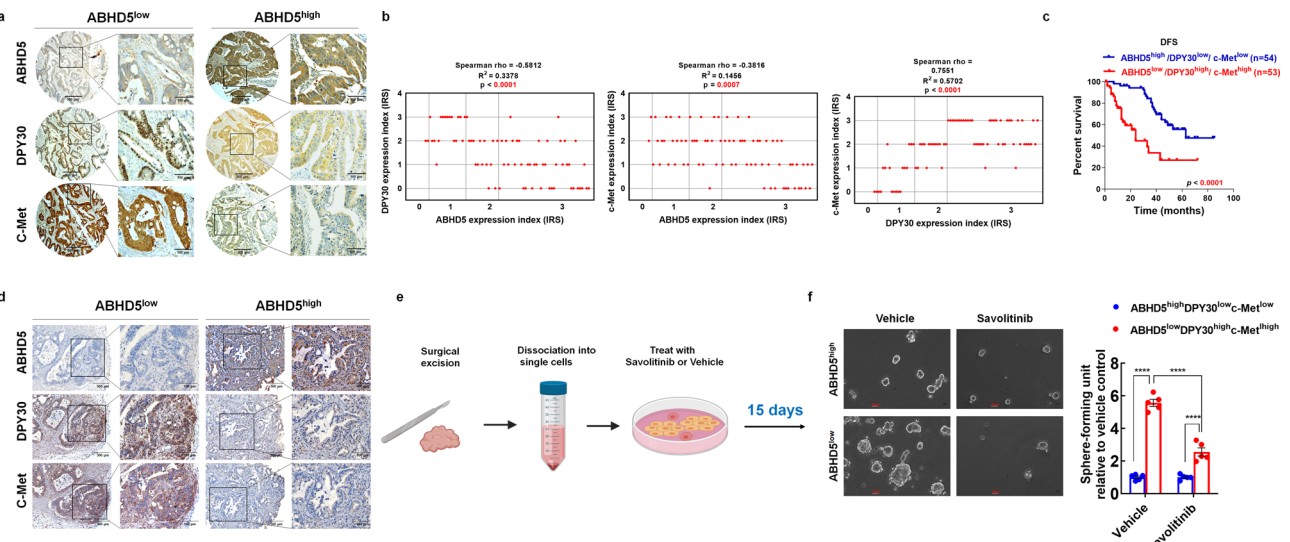

**Fig. 7 ABHD5^low/DPY30^high/c-Met^high CRCs show worse prognosis compared with ABHD5^high/DPY30^low/c-Met^low CRCs but may benefit from Met inhibitor. a** Represent immunohistochemistry stainings of ABHD5/DPY30/c-Met in CRC tissue microarray. Score = 0–1 (low) and score = 2–3 (high) indicate ABHD5, DPY30, and c-Met levels in representative tumour tissues. Scale bars, 500 μm (main micrographs) and 100 μm (insets). **b** Correlation analyses for the correlation between ABHD5, DPY30 and c-Met in CRC tissues. Plotted data are the expression index calculated from immunostaining. Spearman correlation analysis ($n = 76$ biologically independent samples). **c** Disease-free survival (DFS) of CRC patients with ABHD5^low/DPY30^high/ c-Met ^high ($n = 53$) or ABHD5^high/DPY30^low/ c-Met ^low CRC tumours ($n = 54$ biologically independent samples). **d** Represent immunohistochemistry stainings of ABHD5/DPY30/c-Met in surgically resected CRC tissues. Scale bars, 500 μm (main micrographs) and 100 μm (insets). **e** Schematic of the experiment. Surgically resected ABHD5^high/DPY30^low/c-Met^low or ABHD5^low/DPY30^high/c-Met^high CRC tumours were dissociated into single cells. Ex vivo sphere formation assays of tumour-derived cells were performed in the presence of 1 μM savolitinib or DMSO (vehicle). **f** Representative images and statistical analyses of cancer cell spheres derived from surgically resected ABHD5^high/DPY30^low/c-Met^low or ABHD5^low/DPY30^high/c-Met^high CRC tumours ($n = 5$ biologically independent samples) (Data are shown as mean ± s.e.m. The correlation analysis was determined by Spearman correlation coefficient in panel (**b**). Kaplan−Meier survival method was used in panel (**c**). Panel (**f**) was analysed using two-way ANOVA and Sidak's multiple comparison test. ****$p < 0.0001$. Source data are provided as a Source Data file).

To evaluate the therapeutic potential of our in vitro findings, we isolated cancer cells from surgically resected tissues from patients characterized as ABHD5^high DPY30^low c-Met^low or ABHD5^low DPY30^high c-Met^high (Fig. 7d) and assessed sphere formation in the presence of the c-Met inhibitor Savolitinib (Fig. 7e). In this assay, ABHD5^low DPY30^high c-Met^high CRC cells showed significantly greater sphere formation than ABHD5^high DPY30^low c-Met^low CRC cells, and continuous treatment of these patient-derived CRC cells with Savolitinib significantly suppressed sphere formation in the ABHD5^low DPY30^high c-Met^high group but had little effect in the ABHD5^high DPY30^low c-Met^low group (Fig. 7f). Our results strongly imply that therapeutic targeting of c-Met is a potentially effective strategy for eradicating the CRC stem cell compartment, characterized as ABHD5^low DPY30^high c-Met^high.

## Discussion

We previously identified the lipolytic factor *ABHD5* as an important tumour suppressor gene in CRCs and demonstrated that loss of *ABHD5* significantly promotes CRC tumourigenesis and metastasis[15]. However, the underlying molecular mechanism remained incompletely clarified, and no pharmacological therapy specifically targeting ABHD5 pathway was available. It is therefore urgent to reveal the mechanism by which ABHD5 suppressing the development and progression of CRC. Here, we show that loss of ABHD5 activates c-Met to increase and sustain the stemness of CRC cells in a non-canonical and metabolic independent manner and is synthetic lethal with c-Met inhibition. Our results from cell line models, mouse models and patient samples strongly support the following mechanism:

ABHD5 interacts in the cytoplasm with DPY30, an important subunit of the SET1A complex, to promote its ubiquitination and inhibit its nuclear translocation. Loss of ABHD5 increases the nuclear translocation of DPY30, thus promoting SET1A-mediated mono-methylation of YAP at K342, which sequesters YAP in the nucleus. Nuclear YAP consequently binds TEAD to drive the transcription of c-Met, thus promoting c-Met expression and favoring CRC stemness (Fig. 8, working model).

ABHD5 has long been considered a co-factor of PNPLA2. Notably, our series of studies gradually revealed the PNPLA2-independent roles of ABHD5 in the cancer field. We found here that in *PNPLA2* knockout CRC cells, silencing *ABHD5* also promotes the methylation of YAP, further suggesting a PNPLA2-independent effect of ABHD5 on YAP signalling. Intriguingly, we found that *PNPLA2* depletion activates the expression and nuclear translocation of β-Catenin but not YAP. As shown in Supplementary Fig. 8f, the expression of PNPLA2 was significantly increased under *ABHD5* silencing in CRC cells. We therefore cannot exclude the possibility that the modestly decreased expression of β-Catenin in *ABHD5* knockdown cells may partially result from the increased expression of PNPLA2. Therefore, the molecular mechanism responsible for the regulation of β-Catenin under conditions of ABHD5 deficiency may be complicated and needs further exploration. Since we demonstrate a novel role of ABHD5 in regulating the activity of histone methyltransferase complexes, we propose that ABHD5-regulated histone/non-histone methylation is an important mechanism responsible for the distinct phenotype between *Abhd5* and *Pnpla2* knockout mice previously reported. While in this study we focused on the methylation of a non-histone protein, YAP,

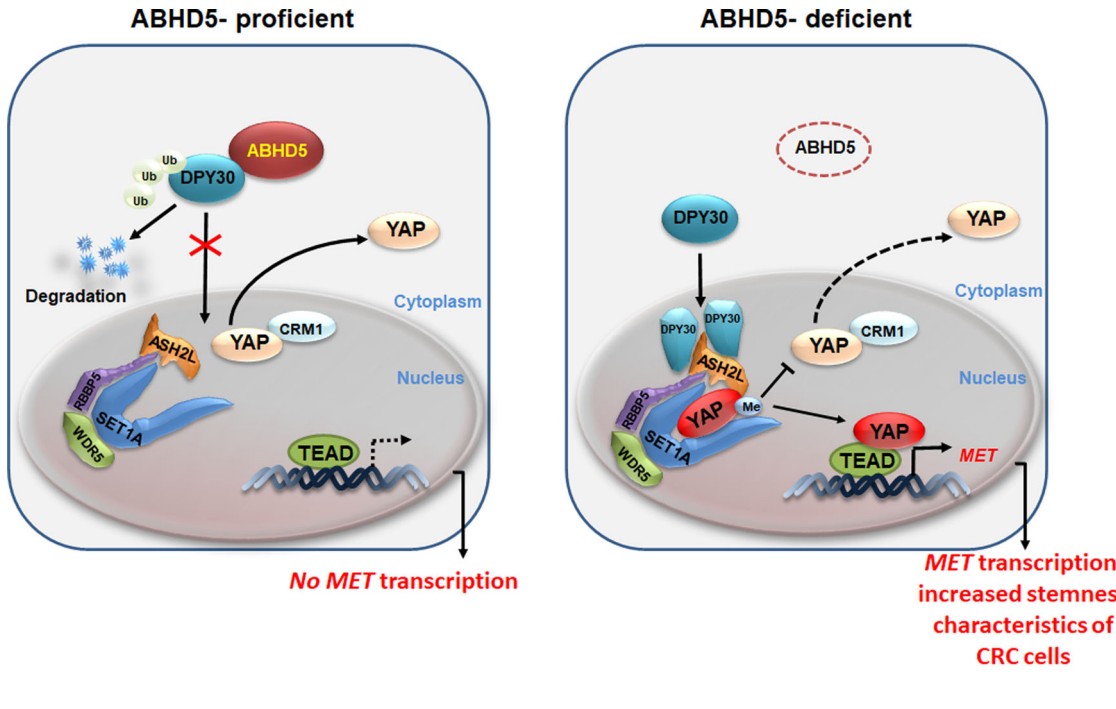

**Fig. 8 Working model.** ABHD5 interacts in the cytoplasm with DPY30, the core subunit of the SET1A complex, to sequester DPY30 in cytoplasm and prevent its nuclear translocation. In ABHD5-deficient CRC cells, loss of ABHD5 releases DPY30 to translocate into the nucleus and achieve SET1-mediated mono-methylation of YAP at K342, which increases the nuclear location of YAP and YAP-induced transcription of c-Met for increasing and sustaining cancer cell stemness, consequently promoting the development and progression of CRCs.

additional studies should expand to the relevance of ABHD5 in histone modification and chromatin remodelling.

YAP activation plays a critical role in promoting the tumourigenesis and metastasis of CRCs[25–31]. Numerous studies indicate that the post-translational modification of YAP is critical for controlling its subcellular localization and activity. Phosphorylation of YAP at S127 by LATS1/2 sequesters it in the cytoplasm for ubiquitination and degradation, while phosphorylation at S128 by NLK induces nuclear localization[45,46]. The virus-activated kinase IKK 3 phosphorylates YAP at S403 and thereby triggers lysosomal degradation[47], and AMPK inhibits YAP activity by phosphorylating S94 in response to energy stress[48]. In addition to phosphorylation, lysine methylation is an important post-translational modification that regulates YAP localization and activation. Mono-methylation of YAP at K494 by SET7 is essential for cytoplasmic retention via an unknown mechanism[49], while mono-methylation at K342 by SET1A sequesters YAP in the nucleus to activate its oncogenic function[33]. We demonstrated that loss of ABHD5 retain YAP in the nucleus via promoting SET1A methylation of YAP, further explaining how YAP methylation and activation are triggered in cancer cells. Although histone lysine methyltransferase complexes primarily reside in the nucleus and target histones, their presence in the cytosol has been suggested[44,50], and several non-histone substrates have been identified[50,51]. DPY30 localizes to the Golgi apparatus and plays a critical role in the endosomal transport of specific cargo proteins[44]. Our findings regarding the interaction between ABHD5 and DPY30 in the cytoplasm suggest several intriguing subjects for future studies: Does ABHD5 also localize to the Golgi? Are DPY30 and ABHD5 components of unidentified methyltransferase complexes in the trans-Golgi network, where they affect endosomal transport by controlling the methylation of a trafficking regulator? In addition, the interaction between

ABHD5 and DPY30 may suggest a new approach for elucidating the molecular function of DPY30 beyond its identity as a conserved H3K4 methyltransferase complexes subunit. Indeed, DPY30 is overexpressed in cancer, and high DPY30 expression levels are correlated with poor prognosis and functionally promote tumourigenesis in certain types of cancer[43]. Moreover, DPY30 is required for the enhanced proliferation, motility and EMT of epithelial ovarian cancer cells because it promotes vimentin expression through H3K4 trimethylation at the vimentin promoter[43]. A recent study found that DPY30 promotes MYC, an oncogene that is deregulated in up to 70% of human cancers and critically associated with cancer metabolic reprogramming[52], and regulates chromatin accessibility to enable the efficient binding of MYC to its genomic targets[53]. These findings strongly suggest an oncogenic role of DPY30, but the mechanism by which DPY30 is regulated in cancer cells remains unknown. Our study identified ABHD5 as an upstream factor that regulates DPY30 degradation and nuclear translocation, suggesting the mechanism underlying DPY30 overexpression and activation in cancer pathogenesis.

In fact, our findings highlight the therapeutic potential of improving the efficacy of c-Met inhibitors by targeting YAP or DPY30, the downstream mediators of the cellular consequences of loss of ABHD5, in patients harbouring CRC with low ABHD5 expression. DPY30 is an upstream agonist of YAP that potentiates the oncogenic activity of YAP through post-translational modification, and thus localization, to activate c-Met transcription; therefore, targeting DPY30 is potentially a more effective and direct approach to overcome the effects of excessive YAP and c-Met expression. Furthermore, DPY30 has certain characteristics of an ideal therapeutic target; for example, its expression level is increased specifically upon the loss of ABHD5. We therefore propose that combination therapy with a DPY30 inhibitor and a

c-Met inhibitor will have a synergistic effect on tumour suppression. Shah et al.[54] have reported the identification of peptides targeting DPY30 with efficacy in certain types of leukaemia; perhaps we can further our studies using these peptides in CRC and/or CSC models.

This study for the first time reveals an unrecognized and non-canonical role of ABHD5 in regulating SET1A-induced YAP and histone methylation, further explaining the molecular mechanism for the tumour suppressor function of ABHD5, and highlighting the metabolic independent pathway by which the aberrant metabolic genes controlling the activation of oncogenic signalling. Further studies focused on the effect of the classic lipolytic factor, ABHD5, on epigenetic modification of histone/non-histone proteins, and its relevance to cancer pathogenesis should be derived from this study.

## Methods

**Creation of Apc$^{Min/+}$ mice lacking intestinal Abhd5**. Intestine-specific *Abhd5*-knockout mice were generated by mating *Abhd5*-floxed mice created by Cyagen Biosciences (Suzhou) Inc. with B6.Cg-Tg (Vil1-cre) 977Gum/J mice (Jackson Laboratory, stock #004586). A male Apc$^{Min/+}$ mouse on the C57BL/6J background was purchased from The Jackson Laboratory (stock #002020) and crossed with female intestine-specific *Abhd5*-knockout mice to produce Apc$^{Min/+}$ mice lacking *Abhd5* in the intestine and their control littermates for experiments. The Institutional Animal Care and Use Committee of Third Military Medical University (Army Medical University) approved all animal procedures.

**Cells**. Detailed information is provided in Supplementary Materials. All cell lines were received from the companies as early passages and were propagated and passaged as adherent cell cultures according to the instructions provided by the ATCC and Fuheng Biology. Cells were maintained in adherent conditions at 37 °C in a humidified atmosphere containing 5% $CO_2$. The medium was replaced three times weekly, and the cells were passaged using 0.05% trypsin/EDTA (Corning) and preserved at early passages.

**Antibodies and reagents**. Detailed information is provided in Supplementary Materials.

**Anchorage-independent growth assay**. 1.2% agar and 2× RPMI 1640 supplemented with 20% FBS were 1:1 mixed, and 1.5 ml of which was layered onto each well of six-well tissue culture plates. 0.7% agar and 2× RPMI 1640 supplemented with 20% FBS were 1:1 mixed, and cells ($5 \times 10^3$) in logarithmic growth phase were added. The cell mixture was added to the top of the 0.6% agar layer. Cells were incubated at 37 °C in 5% $CO_2$ for 14 days, and the media was changed every 3 days. After 14 days in a 37 °C 5% $CO_2$, photographs of soft agar colonies were taken after staining with iodinitrotetrazolium violet (4 mg/ml, Sigma). The number and the size of the colonies were analysed with ImageJ (NIH, Bethesda, MD).

**Sphere formation assay**. Spheres were cultivated in ultralow adherence (ULA) polystyrene plates (Corning), in DMEM/F12 with 20 mg/ml insulin (Sigma I0908), 20 ng/ml EGF (R&D Systems 236-EG), 10 ng/ml bFGF (R&D Systems 233-FB/CF), 3 mg/ml D-glucose (Sigma 49139), and 1% penicillin–streptomycin (PanBiotech P06-07100). Briefly, for the sphere formation assay, the dissociated colon cancer cells remaining after the proliferation assay were stained with 1 mg/ml 7-AAD (Sigma A9400) during 20 min at room temperature, before sorting and seeding 600 viable (7-AAD negative) single cells per well in 24-well ULA plates with a FACSAria (Becton Dickinson), in 500 μl medium per well. After 10 days in a 37 °C 5% $CO_2$, photomicrographs of spheres were taken. The number and the size of the colonies were analysed with ImageJ (NIH, Bethesda, MD).

**Flow cytometry analysis and cell sorting**. HCT116 colon cancer cells were stained using APC-conjugated CD133 (BD Biosciences) and PE-Cy7-conjugated CD44 (BioLegend). Samples were analysed on a BD LSRII flow-cytometer (Bekton Dikinson, Franklin Lakes, NJ, USA). Analysis of cytometric data was performed using FACSDiva software (Bekton Dikinson).

For the identification of ALDH$^+$ cells, the ALDEFLUOR kit was used to sort ALDH$^+$ cells with high ALDH enzymatic activity. Single-cell suspensions were made in ALDH assay buffer containing the ALDH substrate-BAAA (BODIPY-aminoacetaldehyde, 1 mmol/L/$1 \times 10^6$ cells) and then incubated for 40 min at 37 °C. In each experiment, the specific ALDH inhibitor diethylaminobenzaldehyde (DEAB) was used as a control at a concentration of 15 mmol/l. The specific ALDH activity was calculated according to the difference in activity between the presence and absence of the inhibitor DEAB.

**Screening of the FDA-approved drug library**. A 1600-drug FDA-approved library was purchased from Topscience (Shanghai, China). Control and *ABHD5*-knockdown HCT116 cells were seeded into 96-well plates at a density of $1 \times 10^5$ cells/well for 24 h before each drug was added. High-throughput screening was conducted according to the manufacturer's instructions as reported[55].

**Apoptosis assay**. To measure apoptosis by IncuCyte Zoom instrument (Essen BioScience), control and *ABHD5*-knockdown cells were plated on Costar 3596 plates and treated with c-Met inhibitors at different concentrations for 24, 48, and 72 h. Caspase 3 activation was measured with Cell-Player 96-Well Kinetic Caspase 3 reagent (Essen BioScience 4440). Green fluorescent images were acquired every 2 h.

**TOP/FOP Flash dual luciferase assay**. Cells were transfected with the TOPFlash or FOPFlash reporter plasmids, and test plasmids by using Lipofectamine PLUS transfection reagent (Invitrogen). Upon transfection, cells were cultured for 48 and 72 h. Luciferase activity in cell lysates was measured by the luciferase assay system (Promega) in a Berthold Lumat LB 9507 luminometer. We normalized the relative reporter activity to the activity of co-expressed β-galactosidase.

**ChIP assays**. A total of $1 \times 10^7$ cells were prepared for the ChIP assay. The ChIP protocol was performed following the methods reported[56]. Quantification of all ChIP samples was performed by quantitative PCR (qPCR) using a TaKaRa SYBR Premix Ex Taq kit and an ABI 7500 Fast system. The data are presented as the fold changes calculated for each antibody ChIP value (IP/Input, the percentage of input) relative to the IgG control ChIP value.

**Multiomics analyses of Hi-C, ATAC-seq, ChIP-seq and RNA-seq**. Integrated analyses of chromatin conformation (Hi-C), histone methylation (H3K4m3) (ChIP-seq), chromatin accessibility (ATAC-seq) and transcriptomic (RNA-seq) signatures were performed following the methods reported[57].

**Cytosolic and nuclear fractionation**. Cells were homogenized in lysis buffer (50 mM Tris (pH 7.5) 1% Triton X-100, 100 mM NaCl, 10 mM tetrasodium pyrophosphate, 10 mM NaF, 1 mM EDTA, 1 mM NaV, 1 mM EGTA, 1 mM phenylmethylsulphonyl fluoride and 1 μg/ml each aprotinin, leupeptin and pepstatin) followed by gentle sonication on ice. For nucleo-cytoplasmic fractionation, intracellular proteins were prepared using an NE-PER Nucleus and Cytoplasmic Extraction Reagent Kit (Thermo Fisher Scientific, Rockford, IL, USA) according to the manufacturer's protocol. Following determination of protein concentrations, proteins were separated on an SDS-polyacrylamide gel and electrophoretically transferred to a polyvinylidene difluoride (PVDF) membrane, which was then incubated with primary antibody.

**Immunoprecipitation and immunoblotting**. Cells were lysed using a mild lysis buffer. Cell lysates were centrifuged for 10 min at 4 °C, and supernatants were used for immunoprecipitation. Immunoprecipitates were washed four times with lysis buffer, and proteins were eluted with SDS-PAGE sample buffer. Immunoblotting was performed using a standard protocol.

**Human proteome microarray**. The recombinant GST-ABHD5 fusion proteins were labelled with Biotin (Full Moon Biosystems) and used to probe the ProtoArray Human Protein Microarray (Wayen Biotechnologies). The screen and validation of ABHD5-binding proteins were followed the standard procedure as reported[58].

**Xenograft studies**. All animal experiments were approved by the Institutional Animal Care and Use Committee of Third Military Medical University (Army Medical University) in accordance with the Guide for the Care and Use of Laboratory Animals. Six- to eight-week-old NOD/SCID mice were purchased from the Chinese Academy of Sciences Shanghai SLAC Laboratory Animal Co. (SLACCAS, Shanghai, China) and acclimated for 4 days. For subcutaneous tumour, each mouse was injected subcutaneously in the right lateral flank with $1 \times 10^6$ HCT116 cells suspended in 0.2 ml 1:1 matrigel and randomized based on tumour volume, and dosing began when tumours reached 120–275 mm$^3$. The subcutaneous tumour sizes were measured every 3 days in two dimensions with calipers and calculated using the formula $(L \times W^2)/2$, where $L$ is the length and W is the width. For lung metastatic xenografts, each mouse was injected with $1 \times 10^6$ luciferase reporter-expressed HCT116 cells via tail vein, and tumour growth was monitored by bioluminescence imaging using IVIS Spectrum (Perkin-Elmer). After transplantation, cells were allowed to grow for 1 week, and then mice with established tumours were randomly sorted into different treatment groups with 6 mice/group. Savolitinib was formulated in acidic CMC-Na 0.5% (pH = 2.1) and dosed orally once daily at the indicated concentrations, and Verteporfin was formulated in 8% DMSO and dosed intraperitoneally once daily at the indicated concentrations for all studies.

**Human tissue samples**. Tissue chips consisting of human CRC specimens with patient survival follow-up information were purchased from Shanghai Outdo Biotech Co., Ltd, and used specifically for analysis of the associations between ABHD5, DPY30, c-Met and survival. The surgically resected tissues from CRC patients for ex vivo sphere formation assay were collected from Fuling Central Hospital. All human experiments were approved by the Ethics Committee of Fuling Central Hospital and Southwest Hospital.

**Statistics and reproducibility**. All experiments were performed in triplicate at least three independent times. All data are shown as the mean ± s.e.m. Statistical analysis was performed with unpaired two-sided Student's $t$ test or two-way ANOVA with the Sidak correction. Kaplan−Meier survival method was used to evaluate Disease-free survival (DFS). The correlation analysis was determined by Spearman correlation coefficient. A probability value of $p \leq 0.05$ was considered statistically significant. All statistical analyses were performed using GraphPad Prism 8.

**Reporting summary**. Further information on research design is available in the Nature Research Reporting Summary linked to this article.

## Data availability

The processed Microarray data used in this study are available at the gene expression omnibus (GEO) database under accession code GSE185056. The data that support the findings of this study are available within the article and its Supplementary Information files. Because a further study based on the multiomics results of Hi-C, ATAC-seq, ChIP-seq and RNA-seq have not yet been published, so the raw data in this section are available from the corresponding author upon reasonable request. Source data are provided with this paper.

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

## Acknowledgements
We acknowledge the assistance for mass spectrometry analysis from Dr. Yunpeng Zhang and Professor Lujian Liao (Shanghai Key Laboratory of Regulatory Biology, School of Life Sciences, East China Normal University, China), the assistance for multiomics analyses of Hi-C, ATAC-seq, ChIP-seq, and RNA-seq from Biomarker Technologies Corporation (Beijing, China), the assistance for the construction and packaging of adeno-associated virus from Genomeditech (Shanghai, China) and OBiO Technology (Shanghai, China), and the assistance for the construction of Flag-DPY30, wild-type ABHD5 and mutant ABHD5 plasmids from Genechem (Shanghai, China). We also gratefully acknowledge the generous gift of YAP mutant plasmids and anti-YAP K342m antibody from Dr. Lan Fang and Professor Ping Wang (Shanghai Tongji University, China). This work was supported in part by grant numbers 82072745 (J.O.), 81772647 (J.O.), and 81874072 (H.L.) from the National Natural Science Foundation of China, and grant numbers SWH2016ZDCX1002 (H.L.) and SWH2017YBXM-30 (X.Z) from Hospital Management Projects.

## Author contributions
Conceived and designed the experiments: J.O., H.L. Q.Z. and G.X. Performed the experiments: Y.G., Y.C., L.W., S.W., K.S., C.L., Y.D., Ya.Z., Yu.Z., C.Z., W.Z., J.H., Y.W., Y.L., X.Z. Analysed the data: H.W., J.T., K.S., L.W. and J.O. Discussed the data: J.O., H.L. Q.Z. and G.X. Wrote the paper: J.O., H.L., Q.Z. and G.X. All authors read and approved the final submitted manuscript.

## Competing interests
The authors declare no competing interests.
