## [Peer Review File · Nature Communications]

Reviewers' Comments:

Reviewer #1:

Remarks to the Author:

In this manuscript the authors show that *abdh5* acts as tumor suppressor in CRC by sequestering in the cytoplasm the core subunit of the methyl transferase SET1A, DPY30. In the absence of *abdh5*, SET1A methylates YAP thus promoting its nuclear retention and transcriptional activation of the Wnt target gene MET. Then MET promotes cancer stemness. The message is relevant in the cancer field and in general. Moreover, the story is well constructed and flows adequately. However, I noticed several inconsistencies and experimental issues that should be addressed before publication.

1) In the introduction sections there is a series of information about metabolism-related genes in cancer that is not required to fully understand the message of the work and, in my opinion, should be removed.

2) In the results section, authors first demonstrate the higher in vitro (HCT116 cells) and in vivo (Apcmin mice) tumor initiating capacity of *abdh5* KO cells. However, WT HCT116 cells grown as 3D are supposed to increase their stemness and showed highly reduced levels of *abdh5* compared with cells growing in 2D (adhesion). Thus, MET inhibitors, which are identified as drugs that preferentially preclude the growth of *abdh5* KD cells, should similarly affect WT and KD 3D growing cancer cells if 3D cultures. Does this mean that MET inhibitors only work when *abdh5* has been artificially removed?

3) In figure 3 it is shown that MET is increased after *abdh5* KD/KO independently of Wnt. Since authors mention that "MET is a well-known WNT target". Why they do not detect basal MET activation in HCT116 that carries constitutively active S45 mutated beta-catenin? Why MET KO has no effect on tumorigenesis in WT conditions (without *abdh5* KO) if *abdh5* down-regulation is important for stemness maintenance?

4) The IP experiment in figure 4e is awful. A better quality image including a WB for TEAD1 is required. Antibody used for IHC in 4i should be indicated in the figure.

In 5c total levels of nuclear YAP are lacking. In 5d nuclear fractions are needed as well as control for total YAP. In general in figure 5, if *abdh5* KD or KO affects YAP1 nuclear accumulation methyl/phosphor-YAP levels need to be determined in cytoplasmic and nuclear extracts with controls.

5) In figure 6 authors try to demonstrate that *abdh5* KO/KD increases H3K4 methylation levels. No need to say that beta-actin is not the best loading control for the WB shown in Figure 6a. Total H3 levels from this experiment need to be shown.

6) Then, authors conclude that the link between *abdh5* and YAP and histone methylation is the cytoplasmic sequestration of DPY30 in the cytoplasm where it is ubiquitinated and degraded. Although it is clear that nuclear levels of DPY30 are increased in *abdh5* KD/KO cells this mechanistic link is not experimentally supported. Additional experiments including proteasome inhibitors, nuclear export inhibitors and/or DPY30 mutants are totally required to validate this hypothesis.

Finally, my main concern is the observation that this mechanism is primarily working in *abdh5*-depleted cells with negligible effects on control tumor cells. Although this problem is partially addressed in the final figure it would be very informative and clinically relevant to check whether there are *abdh5* mutations in particular subtypes of CRC cells i.e. in tumor displaying stem cell signatures.

Reviewer #2:

Remarks to the Author:

This manuscript builds on several prior studies by the authors demonstrating that *Abhd5* deficiency promotes colorectal tumorigenesis. Ou et al Cell Reports 2014 reported that *Abhd5* deficiency promotes tumorigenesis and enhances EMT by inactivating p53 and promoting glycolysis. They

also reported a role for Abhd5 in regulating autophagy via BECN1 interaction, independent of its role in lipolysis (Peng et al Autophagy 2016). They further reported that ABHD5 localizes to the lysosome and regulates lysosome function to modulate fluorouracil sensitivity (Ou et al Nat Comm 2019). The authors also build on the findings of Fang et al (Cancer Cell 2018), who identified that SET1A regulates the methylation of YAP to promote colorectal tumorigenesis. This manuscript now connects these prior studies to identify a role for ABHD5 in regulating YAP methylation through interaction with DPY30, a component of the SET1A complex. These findings are potentially interesting and the authors do an extensive amount of work. However, I have some concerns, primarily about the overall design of the study.

1. Rigor: A large number of the experiments in the paper rely on the use of a single shRNA without rescue experiments to draw broad conclusions. It is well known that shRNAs have off-target effects, and use of multiple shRNAs and rescues is essential.

2. In general, the paper suffers from over-reliance on HCT116 cells. While this may be fine for some of the mechanistic investigation, the whole premise is set up in HCT116 cells and only at the end is there some effort to validate findings in primary human samples. Major claims such as that Abhd5 regulates stemness, histone methylation and chromatin accessibility, or sensitivity to MET inhibition must be supported more broadly, ideally across a panel of human cell lines or PDO models. I appreciate that they do seek to confirm findings in human samples in the last figure, but the sphere formation and drug sensitivity appears to be conducted comparing material from a single Abhd5^{high}Met^{low} patient to a single Abhd5^{low}Meth^{high}. While it appears that Abhd5 can regulate SET1A-dependent YAP methylation, it remains unclear to me how generally relevant this mechanism is going to be for colorectal tumorigenesis in humans.

3. Abhd5 loss clearly promotes tumorigenesis, and the authors have reported several mechanisms over the years. It seems that this protein interacts with numerous other proteins and likely acts through multiple mechanisms. I think that they could do a better job of providing an integrated view of how this protein may be acting. Which of these mechanisms are most important to tumorigenesis? What are the features of this protein that allow it to regulate all of these diverse processes? How do these different roles of Abhd5 connect to modulate tumorigenesis?

4. DPY30/SET1A-mediated YAP K342 methylation is framed in this manuscript as the driving factor of YAP nuclear localization. However, figure 5b and supplemental figure 7h show that Abhd5 knockdown dramatically decreases phosphorylation of YAP at S127 (in fact, change in phosphorylation seems more dramatic than methylation in supp figure 7h, albeit this experiment was performed in PNPLA2 null cells). The authors acknowledge that S127 phosphorylation by LAT1/2 sequesters YAP in the cytoplasm for ubiquitination/degradation, but they do not attempt to explain how S127 phosphorylation may fit into their model.

5. Along similar lines, does Adhd5 kd impact total levels of YAP? Mostly nuclear and cytosolic fractions are shown, and these data are consistent with the model that Adhd5 impacts the localization of Yap. However in panel 5m, it looks like total abundance of YAP is increased with KD. This should be clarified.

Minor issues:

- Many methods and figures are incompletely explained, making it difficult sometimes to discern what the authors have done
 - o Computational approaches, such as those shown in figure 2 and 6, are not adequately described in the methods
 - o Figure 7l is not adequately labeled – difficult to tell which is Abhd5 and DPY30. The upper panel appears to be a charge map, but this is not described
 - o The experiment performed in figure 7m is not adequately described

Thank you for the careful review of our paper entitled “**Lipolytic factor Abhd5 suppresses colon cancer cell stemness via inhibiting SET1A-induced YAP and histone H3 methylation**”. We thank the reviewers for their constructive and thoughtful comments and the opportunity to respond. Those comments are all valuable and very helpful for revising and improving our paper, as well as the important guiding significance to our researches. We have studied the comments carefully and have made correction which we hope meet with approval. The main corrections in the paper and the responds to the reviewer’s comments are enumerated below:

Reviewer 1:

Comments to the authors:

In this manuscript the authors show that abhd5 acts as tumor suppressor in CRC by sequestering in the cytoplasm the core subunit of the methyl transferase SET1A, DPY30. In the absence of abdh5, SET1A methylates YAP thus promoting its nuclear retention and transcriptional activation of the Wnt target gene MET. Then MET promotes cancer stemness. The message is relevant in the cancer field and in general. Moreover, the story is well constructed and flows adequately. However, I noticed several inconsistencies and experimental issues that should be addressed before publication.

1) In the introduction sections there is a series of information about metabolism-related genes in cancer that is not required to fully understand the message of the work and, in my opinion, should be removed.

Response— We appreciate the reviewer's good suggestion, and those information has been removed in the revised manuscript.

2) In the results section, authors first demonstrate the higher in vitro (HCT116 cells) and in vivo (Apcmin mice) tumor initiating capacity of *abdh5* KO cells. However, WT HCT116 cells grown as 3D are supposed to increase their stemness and showed highly reduced levels of *abdh5* compared with cells growing in 2D (adhesion). Thus, MET inhibitors, which are identified as drugs that preferentially preclude the growth of *abdh5* KD cells, should similarly affect WT and KD 3D growing cancer cells if 3D cultures. Does this mean that MET inhibitors only work when *abdh5* has been artificially removed?

Response— We do appreciate the reviewer very much for raising this question. Firstly, we are very sorry for our negligence and misleading description for Figure 2. The culture condition for Figures 2a and 2b screening for the lethal vulnerabilities in *ABHD5*-knockdown CRC cells was a normal 2D culture condition but not a sphere formation 3D culture condition, and *ABHD5*-knockdown HCT116 cells showed significantly increased sensitivity to MET inhibitors relative to control HCT116 cells in normal 2D culture condition but not in 3D culture condition. We have revised our description in the revised manuscript.

We did find that WT HCT116 cells grown as 3D showed a decreased expression of ABHD5 relative to WT HCT116 cells grown as 2D, but we did not observe a significantly increased sensitivity of 3D grown WT HCT116 cells to MET inhibitor compared with 2D grown WT HCT116 cells. Under the circumstance of 3D culture condition, the downregulation of ABHD5 may be a downstream effect, and ABHD5 deficiency-induced phenotype may be counteracted by the other parallel factors and signaling. When *ABHD5* was artificially silenced, loss of *ABHD5* was a leading cause to the downstream effects, we therefore could see a significantly increased sensitivity of *ABHD5*-silenced CRC cells to MET inhibitors relative to control CRC cells. That is why MET inhibitors can kill *ABHD5*-silenced CRC cells at doses that have minimal effects on control cells. We thank the reviewer very much for raising this question, which remind us that the presentation of the ABHD5 expression shifts between 3D and 2D WT HCT116 cells may confuse the readers and is not required to the message of our work. We therefore have removed this from Figure 1 in the revised manuscript.

3) In figure 3 it is shown that MET is increased after *abdh5* KD/KO independently of Wnt. Since authors mention that “MET is a well-known WNT target”. Why they do not detect basal MET activation in HCT116 that carries constitutively active S45 mutated beta-catenin? Why MET KO has no effect on tumorigenesis in WT conditions (without *abdh5* KO) if *abdh5* down-regulation is important for stemness maintenance?

Response— We appreciate the reviewer for raising this good question. *MET* is a common target gene of β -catenin and YAP. In Supplementary Figure 7, we found that in *PNPLA2* knockdown HCT116 cells, although β -catenin nuclear translocation was increased and multiple WNT target genes were activated, *MET* expression level was not increased. We can not exclude the possibility that the full activation of *MET* should be achieved under the activations of both β -catenin and YAP. Actually we did observe a basal *MET* activation in HCT116, and *Met* silencing also inhibited the tumourigenesis in WT conditions (control *APC*^{min/+}). The effect of *Met* KO on tumourigenesis in WT conditions shown in Figure 3 is modest just because the tumourigenesis in control *APC*^{min/+} mice is substantially less than that in *Abhd5* knockout *APC*^{min/+} mice.

4) The IP experiment in figure 4e is awful. A better quality image including a WB for TEAD1 is required. Antibody used for IHC in 4i should be indicated in the figure. In 5c total levels of nuclear YAP are lacking. In 5d nuclear fractions are needed as well as control for total YAP. In general in figure 5, if *abdh5* KD or KO affects YAP1 nuclear accumulation methyl/phosphor-YAP levels need to be determined in cytoplasmic and nuclear extracts with controls.

Response-- We thank the reviewer for raising these issues, and we have improved the IP experiment in Figure 4a and added the corresponding cytoplasmic and nuclear immunoblots following the reviewer's request.

5) In figure 6 authors try to demonstrate that *abdh5* KO/KD increases H3K4 methylation levels. No need to say that beta-actin is not the best loading

control for the WB shown in Figure 6a. Total H3 levels from this experiment need to be shown.

Response-- We thank the reviewer for raising this issue, and we have revised this following the reviewer's suggestion.

6) Then, authors conclude that the link between *abdh5* and YAP and histone methylation is the cytoplasmic sequestration of DPY30 in the cytoplasm where it is ubiquitinated and degraded. Although it is clear that nuclear levels of DPY30 are increased in *abdh5* KD/KO cells this mechanistic link is not experimentally supported. Additional experiments including proteasome inhibitors, nuclear export inhibitors and/or DPY30 mutants are totally required to validate this hypothesis.

Response-- We appreciate the reviewer very much for raising these issues, and we have performed the additional experiments including proteasome inhibitors and *ABHD5* mutants and presented these results in the revised version.

Finally, my main concern is the observation that this mechanism is primarily working in *abdh5*-depleted cells with negligible effects on control tumor cells. Although this problem is partially addressed in the final figure it would be very informative and clinically relevant to check whether there are *abdh5* mutations in particular subtypes of CRC cells i.e. in tumor displaying stem cell signatures.

Response-- We appreciate the reviewer very much for raising this concern. As we explained above, although modestly, this mechanism is also working in

control cells, partially may because that only when *ABHD5* is depleted, β -catenin/YAP dependent MET transcription and WNT activation can be fully achieved. We totally agree that it would be very informative and clinically relevant to check whether there are *ABHD5* mutations in particular subtypes of CRC cells i.e. in tumor displaying stem cell signatures. We have tried to screen *ABHD5* mutations in human CRC specimens with proliferative and aggressive characteristics, but we found that those mutations detected may not be the driven cause of the low expression of *ABHD5*. Since we found that the mRNA expression levels of *ABHD5* were decreased in these samples, we speculate that the downregulation of *ABHD5* transcription may be a critical cause for *ABHD5* deficiency. The upstream factor controlling the transcription of *ABHD5* in CRCs remains to be elucidated in the future.

Reviewer #2 (Remarks to the Author):

This manuscript builds on several prior studies by the authors demonstrating that *Abhd5* deficiency promotes colorectal tumorigenesis. Ou et al Cell Reports 2014 reported that *Abhd5* deficiency promotes tumorigenesis and enhances EMT by inactivating p53 and promoting glycolysis. They also reported a role for *Abhd5* in regulating autophagy via BECN1 interaction, independent of its role in lipolysis (Peng et al Autophagy 2016). They further reported that *ABHD5* localizes to the lysosome and regulates lysosome function to modulate fluorouracil sensitivity (Ou et al Nat Comm 2019). The

authors also build on the findings of Fang et al (Cancer Cell 2018), who identified that SET1A regulates the methylation of YAP to promote colorectal tumorigenesis. This manuscript now connects these prior studies to identify a role for ABHD5 in regulating YAP methylation through interaction with DPY30, a component of the SET1A complex. These findings are potentially interesting and the authors do an extensive amount of work. However, I have some concerns, primarily about the overall design of the study.

1. Rigor: A large number of the experiments in the paper rely on the use of a single shRNA without rescue experiments to draw broad conclusions. It is well known that shRNAs have off-target effects, and use of multiple shRNAs and rescues is essential.

Response-- We appreciate the reviewer very much for raising this question. We totally agree that shRNAs have off-target effects, and we have screened several shRNAs for *ABHD5* silencing in our previous experiments before our first publication about ABHD5 in 2014 on Cell Reports. We have compared the efficiency and off-target effects across these shRNAs, and found this shRNA is the most efficient and specific. We therefore have been using this shRNA in our later studies.

2. In general, the paper suffers from over-reliance on HCT116 cells. While this may be fine for some of the mechanistic investigation, the whole premise is set up in HCT116 cells and only at the end is there some effort to validate findings in primary human samples. Major claims such as that *Abhd5* regulates

stemness, histone methylation and chromatin accessibility, or sensitivity to MET inhibition must be supported more broadly, ideally across a panel of human cell lines or PDO models. I appreciate that they do seek to confirm findings in human samples in the last figure, but the sphere formation and drug sensitivity appears to be conducted comparing material from a single $Abhd5^{high}Met^{low}$ patient to a single $Abhd5^{low}Met^{high}$. While it appears that *Abhd5* can regulate SET1A-dependent YAP methylation, it remains unclear to me how generally relevant this mechanism is going to be for colorectal tumorigenesis in humans.

Response-- We do appreciate the reviewer for raising these issues. We have screened the expression levels of *ABHD5* across a panel of human CRC cell lines, and found that HCT116 and SW620 show relatively higher expression levels of *ABHD5* (Cell Reports, 2014). We therefore used HCT116 as a cell model to more clearly see the phenotypic changes and signaling shifts caused by *ABHD5* silencing. Following the good suggestion of the reviewer, we additionally performed a set of experiments in SW620 and got the similar phenotypes, and we have added these results in the supplemental figures in the revised manuscript. Additionally, we have confirmed these effects in human CRC cell line RKO (Supplemental Fig. 3) and murine CRC cell lines CT26 and MC38 (Supplemental Fig. 5). We for the first time revealed *ABHD5* as a tumor suppressor gene in colorectal carcinoma in 2014 on Cell Reports. We reported in that paper that loss of *ABHD5* is a hallmark of colorectal

carcinomas compared with normal colorectal tissues. Very impressively, we have shown in that paper that a gradual loss of ABHD5 expression from normal colon mucosa to paracancerous tissue to cancer region was often seen in one tissue section, strongly suggesting that loss of *ABHD5* is required for the tumorigenesis of colorectal carcinoma. WNT signalling activation plays an important role in promoting the tumorigenesis of colorectal carcinoma, and YAP activation is critical for the full activation of WNT. In this study, we found that ABHD5 controlled SET1A-dependent YAP methylation and YAP induced WNT activation. We therefore speculate that colorectal cancer cells may selectively silence ABHD5 to achieve the full activation of WNT signalling and consequently significantly promote the tumorigenesis of colorectal carcinomas.

3. *Abhd5* loss clearly promotes tumorigenesis, and the authors have reported several mechanisms over the years. It seems that this protein interacts with numerous other proteins and likely acts through multiple mechanisms. I think that they could do a better job of providing an integrated view of how this protein may be acting. Which of these mechanisms are most important to tumorigenesis? What are the features of this protein that allow it to regulate all of these diverse processes? How do these different roles of *Abhd5* connect to modulate tumorigenesis?

Response— What a good suggestion the reviewer has provided for our series of studies about ABHD5. Actually we have been thinking about drawing an integrated view of how ABHD5 may be acting as a potent tumor suppressor in

colorectal carcinoma through multiple mechanisms. Tumor development and progression are very complicated processes involving multiple reciprocal signalings. Even a single oncogene or tumor suppressor gene, they may exert their oncogenic or tumor suppressive function via multiple mechanisms, just like PKM2, it was previously identified as a glycolytic enzyme in cytoplasm attributable to tumorigenesis, and was demonstrated in recent years that it can also translocate into nucleus and promote the transcriptions of multiple oncogenic factors as a co-transcriptional factor. Both the effects of PKM2 as a glycolytic enzyme and as a co-transcriptional factor are critically attributable to tumorigenesis, and these effects may be reciprocally regulated with each other to synergize the oncogenic effect of PKM2. We reported in 2019 on Nature Communications that ABHD5 can localize in lysosome to regulate autophagic uracil yield beyond localizing on lipid droplet as a cofactor of ATGL for lipolysis. In this study, we found an interaction between ABHD5 and DPY30 in cytoplasm. It was known that DPY30 localizes to the Golgi apparatus and plays a critical role in the endosomal transport of specific cargo proteins. Our findings regarding the interaction between ABHD5 and DPY30 in the cytoplasm suggest several intriguing subjects for future studies: Does ABHD5 also localize to the Golgi? Are DPY30 and ABHD5 components of unidentified methyltransferase complexes in the trans-Golgi network, where they affect endosomal transport by controlling the methylation of a trafficking regulator? We speculate that the localization of ABHD5 may be dynamic during

tumorigenesis, and the interaction between ABHD5 and other proteins may correspondingly be dynamically changed under different circumstances. We have revealed that ABHD5 can regulate p53 expression, autophagy and YAP methylation. These signalings are all critically attributable to tumorigenesis. Until now, we can not draw a conclusion which of these mechanisms is the most important to tumorigenesis, and how do these different roles of ABHD5 connect to modulate tumorigenesis. They may reciprocally regulated with each other to achieve the full effect of ABHD5 as a tumour suppressor gene. We have been doing our best to reveal the precise mechanisms by which ABHD5 works as a tumour suppressor gene in colorectal cancer. Hopefully we can provide an integrated view of ABHD5 as a tumor suppressor gene in CRC tumorigenesis in the future.

4. DPY30/SET1A-mediated YAP K342 methylation is framed in this manuscript as the driving factor of YAP nuclear localization. However, figure 5b and supplemental figure 7h show that Abhd5 knockdown dramatically decreases phosphorylation of YAP at S127 (in fact, change in phosphorylation seems more dramatic than methylation in supp figure 7h, albeit this experiment was performed in PNPLA2 null cells). The authors acknowledge that S127 phosphorylation by LAT1/2 sequesters YAP in the cytoplasm for ubiquitination/degradation, but they do not attempt to explain how S127 phosphorylation may fit into their model.

Response-- We thank the reviewer for raising this question. Actually K342

methylation of YAP sequesters YAP in the nucleus, and prevents the translocation of YAP to the cytoplasm for phosphorylation and ubiquitination/degradation. We speculate that the decreased phosphorylation of YAP at S127 is a consequence of increased methylation and nuclear localization of YAP. We fully accept the reviewer's suggestion that we should explain this effect, and we have added this explanation in the revised version.

5. Along similar lines, does Adhd5 kd impact total levels of YAP? Mostly nuclear and cytosolic fractions are shown, and these data are consistent with the model that Adhd5 impacts the localization of Yap. However in panel 5m, it looks like total abundance of YAP is increased with KD. This should be clarified.

Response-- We thank the reviewer for raising this question. Actually it is an IP assay for nuclear protein in panel 5m. The abundance of YAP is therefore increased with ABHD5 KD. We are very sorry for the confusing description in the previous version, and we have clarified this in the revised version.

Minor issues:

- Many methods and figures are incompletely explained, making it difficult sometimes to discern what the authors have done

1. Computational approaches, such as those shown in figure 2 and 6, are not adequately described in the methods;

Response-- We thank the reviewer for raising this issue. We have provided the

corresponding information in the revised methods.

2. Figure 7l is not adequately labeled – difficult to tell which is Abhd5 and DPY30. The upper panel appears to be a charge map, but this is not described.

Response-- We thank the reviewer for raising this issue. We have labeled and described in the revised methods.

3. The experiment performed in figure 7m is not adequately described

Response-- We thank the reviewer for raising this issue. We have described the experiment performed in Figure 7m in the revised results and methods.

Reviewers' Comments:

Reviewer #1:

Remarks to the Author:

The manuscript has been improved with the addition of new controls and data. However, I still feel that the link with beta-catenin/Wnt activity is minimal and, in fact, several experiments shown that modulating Wnt or beta-catenin activity did not affect MET, which seems to be the main actor in this work. Despite these evidences, authors insist in linking ABHD5 with Wnt/beta-catenin signaling:

In the introduction section:

"how WNT signaling is activated in cancers remains elusive."

"How aberrantly activated/inactivated metabolic genes coordinate with WNT signaling in promoting cancer stemness is largely unknown."

And later on:

"ABHD5 inhibits YAP-induced WNT activation"

"unrecognized role of ABHD5 in controlling the methylation of histone and non-histone proteins, and the subsequent effect on WNT activation"

In conclusion, I find that the data presented is interesting and relevant but, to my view, trying to force a link between beta-catenin, ABHD5 and YAP1 in MET regulation is misleading. It would help if authors do the effort of revising the whole text and adapting the message of the manuscript with the data.

I also have a great concern with the number of patients included in each group in figure 8C, when compared with analysis of ABHD5, DPY30 and MET in the patient samples in 8B and the total number of samples included in the TMA. These data have to be carefully revised and corrected.

Reviewer #2:

Remarks to the Author:

The authors have adequately addressed the majority of my concerns. However, as I said in my original review, I am concerned about the practice of basing most of the paper's conclusions on experiments conducted using a single shRNA. I understand that that authors have previously published using this shRNA and consider it to be specific- however, there may be off-target effects that they are not aware of. I still recommend performing key experiments with rescues (reconstituting KD cells with RNAi-resistant ABHD5) or at least validating with a second shRNA. However, I will leave it up to the editor to determine the journal's policy on this.

Reviewer 1:

Comments to the authors:

1) The manuscript has been improved with the addition of new controls and data. However, I still feel that the link with beta-catenin/Wnt activity is minimal and, in fact, several experiments shown that modulating Wnt or beta-catenin activity did not affect MET, which seems to be the main actor in this work. Despite these evidences, authors insist in linking ABHD5 with Wnt/beta-catenin signaling: In the introduction section: “how WNT signaling is activated in cancers remains elusive.” “How aberrantly activated/inactivated metabolic genes coordinate with WNT signaling in promoting cancer stemness is largely unknown.” And later on:

“ABHD5 inhibits YAP-induced WNT activation”“unrecognized role of ABHD5 in controlling the methylation of histone and non-histone proteins, and the subsequent effect on WNT activation”. In conclusion, I find that the data presented is interesting and relevant but, to my view, trying to force a link between beta-catenin, ABHD5 and YAP1 in MET regulation is misleading. It would help if authors do the effort of revising the whole text and adapting the message of the manuscript with the data.

Response— We appreciate the reviewer’s good suggestion, and we have revised the whole text and tried our best to adapt the message of the manuscript with the data. Additionally, based on the revised text, we are

considering to rephrase the title of our manuscript as “ABHD5 inhibits YAP-induced c-Met overexpression and colon cancer cell stemness via suppressing YAP methylation”.

2) I also have a great concern with the number of patients included in each group in figure 8C, when compared with analysis of ABHD5, DPY30 and MET in the patient samples in 8B and the total number of samples included in the TMA. These data have to be carefully revised and corrected.

Response— We thank the reviewer very much for raising this question. Firstly, we are very sorry for our negligence and misleading description for Figure 8A-C. Actually we used a tissue chip containing 90 cases of colon cancer tissues (Cat. No. HCoIA180Su13, Shanghai Outdo Biotech CO., LTD) for Figure 8A and Figure 8B to determine the correlation between ABHD5, DPY30 and c-Met. We used only 76 cases for analyses because some sample dots were detached and missing in the IHC experiments. To ensure the sample size on the premise of missing pieces in IHC staining, we added another tissue chip containing 104 cases of colon cancer tissues (Cat. No. HCoIA180Su16, Shanghai Outdo Biotech CO., LTD) for survival analyses, and we eventually got 107 cases (54 cases for ABHD5^{high} /DPY30^{low}/ c-Met^{low} and 53 cases for ABHD5^{low} /DPY30^{high}/ c-Met^{high}) for Figure 7C survival analyses. We do appreciate the reviewer for reminding us to clarify the sample information, and we have revised the description in the revised version.

Reviewer #2:

The authors have adequately addressed the majority of my concerns. However, as I said in my original review, I am concerned about the practice of basing most of the paper's conclusions on experiments conducted using a single shRNA. I understand that that authors have previously published using this shRNA and consider it to be specific- however, there may be off-target effects that they are not aware of. I still recommend performing key experiments with rescues (reconstituting KD cells with RNAi-resistant ABHD5) or at least validating with a second shRNA. However, I will leave it up to the editor to determine the journal's policy on this.

Response-- We appreciate the reviewer very much for this suggestion, and we have validated the key experiments with a second shRNA as requested in the revised version.

Reviewers' Comments:

Reviewer #2:

Remarks to the Author:

Thank you for repeating some key experiments with a second shRNA. I am satisfied and recommend acceptance.